# Size-resolved particle number emissions in Beijing determined from measured particle size distributions

Jenni Kontkanen[1,2], Chenjuan Deng[3], Yueyun Fu[3], Lubna Dada[2], Ying Zhou[1], Jing Cai[2], Kaspar R. Daellenbach[2], Simo Hakala[2], Tom V. Kokkonen[2,4], Zhuohui Lin[1], Yongchun Liu[1], Yonghong Wang[2], Chao Yan[1,2], Tuukka Petäjä[2], Jingkun Jiang[3], Markku Kulmala[1,2], and Pauli Paasonen[1,2]

[1]Aerosol and Haze Laboratory, Beijing Advanced Innovation Center for Soft Matter Science and Engineering, Beijing University of Chemical Technology, Beijing, China
[2]Institute for Atmospheric and Earth System Research / Physics, Faculty of Science, University of Helsinki, Helsinki, Finland
[3]State Key Joint Laboratory of Environment Simulation and Pollution Control, School of Environment, Tsinghua University, Beijing, China
[4]Joint International Research Laboratory of Atmospheric and Earth System Sciences, School of Atmospheric Sciences, Nanjing University, Nanjing, China

*Correspondence to*: Jenni Kontkanen (jenni.kontkanen@helsinki.fi)

**Abstract.** The climate and air quality effects of aerosol particles depend on the number and size of the particles. In urban environments, a large fraction of aerosol particles originates from anthropogenic emissions. To evaluate the effects of different pollution sources on air quality, knowledge of size distributions of particle number emissions is needed. Here we introduce a novel method for determining size-resolved particle number emissions, based on measured particle size distributions. We apply our method to data measured in Beijing, China, to determine the number size distribution of emitted particles in diameter range from 2 to 1000 nm. The observed particle number emissions are dominated by emissions of particles smaller than 30 nm. Our results suggest that traffic is the major source of particle number emissions with the highest emissions observed for particles around 10 nm during rush hours. At sizes below 6 nm, clustering of atmospheric vapors contributes to calculated emissions. The comparison between our calculated emissions and those estimated with an integrated assessment model GAINS shows that our method yields clearly higher particle emissions at sizes below 60 nm, but at sizes above that the two methods agree well. Overall, our method is proven to be a useful tool for gaining new knowledge of the size distributions of particle number emissions in urban environments, and for validating emission inventories and models. In the future, the method will be developed by modeling the transport of particles from different sources to obtain more accurate estimates of particle number emissions.

## 1 Introduction

Atmospheric aerosol particles have significant effects on climate and air quality, which depend largely on the number and mass size distributions of particles (Stocker et al., 2013; WHO, 2016). Epidemiological studies have shown that long-term exposure to high mass concentrations of particles, especially those with diameters less than 2.5 μm ($PM_{2.5}$), is connected to increased mortality (Lelieveld et al., 2015; Pope and Dockery, 2006). On the other hand, clinical and toxicological studies indicate that ultrafine particles, which have diameters less than 0.1 μm, can have more adverse health effects relative to their mass than larger particles (Donaldson et al., 2005; Maher et al., 2016; Oberdörster, 2001). The premature mortality due to particulate pollution is highest in highly urbanized regions, such as Asian megacities (Lelieveld et al., 2015). In this study, we focus on Beijing, where annual premature deaths attributed to $PM_{2.5}$ were estimated to be approx. 19 000 for the year 2015 (Maji et al., 2018).

High particulate pollution levels in Beijing result from both large emissions of primary particles and production of secondary particles. In Beijing, primary particles are emitted from sources including traffic, cooking activities, fossil fuel combustion and biomass burning (Hu et al., 2017; Liu et al., 2017; Sun et al., 2013; Wang et al., 2020). The relative strength of these sources varies seasonally; for example, coal combustion is a significant source only during the residential heating period (Hu et al., 2017), which is usually between mid-November and mid-March. Secondary particles are produced in atmospheric new particle formation (NPF), which includes the formation of nanometer-sized particles by clustering of atmospheric vapors, and the following growth of particles to larger sizes (Kulmala et al., 2014). Frequent NPF events with high particle formation rates have been observed in Beijing (Chu et al. 2019 and references therein) and they have been suggested to contribute to the formation of haze (Guo et al., 2014).

To implement efficient pollution control strategies in Beijing and other megacities, more knowledge of the size-resolved particle number emissions and their sources is needed. Recently, Cai et al. (2020) applied PMF (Positive Matrix Factorization) analysis to particle size distribution and chemical composition data measured in Beijing to investigate particle emissions from different sources. They used data from April to July 2018, excluding NPF event days from the analysis. They found that particle size distribution between 20 and 680 nm can be described by five factors, including two traffic-related factors, one cooking-related factor and two regional secondary aerosol formation-related factors. The first traffic-related factor had a geometric mean diameter (GMD) of ~20 nm, and it was attributed to emissions from gasoline vehicles. The second traffic-related factor had a GMD of ~100 nm and it was connected to diesel vehicle emissions. The cooking-related factor had a GMD of ~50 nm. The two factors related to regional secondary aerosol formation had bimodal distributions with the main peaks at ~200 nm and ~400 nm. When comparing the contributions of different PMF factors, traffic-related factors explained 44% of particle concentrations between 20 and 680 nm, cooking-related factor 32% and secondary aerosol formation-related factors 24%. The findings of Cai et al. (2020) are in line with other studies applying PMF to particle size distribution data from Beijing (Liu et al., 2017; Wang et al., 2013). The contribution of NPF to particle number concentrations was not separately investigated in any of these studies.

The results of the PMF analysis on traffic-related particle size distributions are consistent with direct measurements of size distributions of traffic-originated particles (Rönkkö and Timonen, 2019). Studies suggest that the size distribution of hot and undiluted motor vehicle exhaust typically contains a mode of non-volatile particles smaller than 10 nm (core mode) and the larger mode (soot mode) with diameters between 30 and 100 nm (Harris and Maricq, 2001; Rönkkö et al., 2007). When exhaust is diluted and cooled in the atmosphere, gaseous compounds in the exhaust can form new nucleation mode particles and condense on core and soot mode particles (Charron and Harrison, 2003; Rönkkö et al., 2007). It was recently shown that dilution and cooling of exhaust also produces significant concentrations of particles smaller than 3 nm (Rönkkö et al., 2017).

Emission inventories are used for understanding the contributions of different regional pollutant sources to concentrations of gaseous and particulate pollutants. The emission inventories are typically based on experimentally determined pollutant emission factors (unit of pollutant emitted per unit of activity) and estimated activity levels (unit of activity per unit of time) for different anthropogenic activities. By adding future scenarios for activity levels and determining emission factors for emerging technologies, it is possible to estimate the impacts of planned emission regulations or other future changes on the emissions. Such emission scenario models can be coupled with atmospheric transport models for integrated assessment modelling of health and climate impacts of planned systemic changes. The integrated assessment model GAINS (Greenhous gas and air pollution interactions and synergies; Amann et al., 2013) has been applied for developing actions for improving air quality in the EU and other parts of the world. Recently, size-segregated particle number emission factors were added to the GAINS model (Paasonen et al., 2016), which makes it possible to also estimate regional particle number emissions and their future development. The first implementation of GAINS particle number emissions to a global Earth system model resulted in particle number concentrations closer to the observations than with the previously used emission inventories (Xausa et al., 2018).

The estimated emissions of gaseous pollutants and particulate matter (PM$_{2.5}$) from integrated assessment models have been found to produce reasonable concentrations in China on regional scale (Wang et al., 2011) and the spatial resolution of the models can be improved to study smaller areas, such as the Beijing-Tianjin-Hebei region (Xing et al., 2017). However, using integrated assessment models to estimate the size distributions of particle number emissions is more challenging. This is because it is laborious to model different processes impacting particle number size distributions, such as coagulation scavenging of small particles, atmospheric NPF, condensational growth of particles, and the possible evaporation of particles emitted from anthropogenic sources (Harrison et al., 2016). There are also gaps in our understanding of several of these processes. A good agreement may be found when directly comparing the observed particle number size distributions to those obtained with an integrated assessment model, but the reasons can be wrong. For example, underestimated anthropogenic emissions may be compensated by overestimated NPF. In order to adequately estimate the contributions of different sources to urban particle number size distributions, it is crucial to develop methods based on ambient observations for determining the size distribution of emitted particles. Besides validating integrated assessment models, observation-based methods can be

directly used to derive particle number emission factors for traffic (see e.g. Mårtensson et al., 2006), needed in different air quality modeling applications.

In this study, we develop and apply a new method for determining size-resolved particle number emissions, based on measured number size distributions of atmospheric particles. First, we describe the scientific basis of the method and discuss the limitations of the method. Then, we apply the method to measurements performed in Beijing, China, during January 2018 –

March 2019, to investigate the size distribution of particle number emissions and its diurnal cycle in this Chinese megacity. We also assess how well emissions determined with our method agree with emissions from the GAINS model.

## 2 Methods

### 2.1 Balance equation for estimating particle number emissions

Population balance equations, derived from aerosol general dynamic equation, have been used to estimate particle formation
rates (Cai and Jiang, 2017; Kulmala et al., 2012), particle growth rates (Kuang et al., 2012), and the effect of transport on aerosol particle size distribution (Cai et al., 2018). In this study, we use the population balance method to estimate particle number emissions into a column extending from the ground to the top of the atmospheric mixing layer (ML). The time-evolution of particle number concentration in size bin $i$ ($N_i$) in this column can be described as

$$\frac{d}{dt}(N_i \times MLH) = E_i + J_{GR_{in},i} - J_{GR_{out},i} - S_{coag,i} - S_{depos,i} . \quad (1)$$


Here $E_i$ (in units of m$^{-2}$ s$^{-1}$) represents emission to the size bin $i$ and $J_{GR_{in},i}$ and $J_{GR_{out},i}$ describe the growth into and out of the size bin $i$. $S_{coag,i}$ and $S_{depos,i}$ describe the losses of particles in the size bin $i$ due to coagulation and deposition. The time derivative of the column number concentration can be divided to two terms: the first one is $\frac{dN_i}{dt} \times MLH$, which describes the change of the column particle number concentration due to processes affecting directly particle number concentration $N_i$, and

the second term is $N_i \frac{dMLH}{dt}$, which describes the dilution of the concentration $N_i$, due to increase of mixing layer height (MLH) in the morning.

By reorganizing Eq. (1) and writing out all the terms, emission $E_i$ is obtained from

$$E_i = \frac{dN_i}{dt} \times MLH - MLH \times GR_{in,i} \times \frac{N_{GR_{in},i}}{\Delta D_{p,GR_{in},i}} + MLH \times GR_{out,i} \times \frac{N_{GR_{out},i}}{\Delta D_{p,GR_{out},i}} + MLH \times CoagS_i \times N_i + MLH \times DR_i \times N_i + N_i \frac{dMLH}{dt}. \quad (2)$$

Here $N_i$ is the number concentration of particles in the size bin $i$. $GR_{in,i}$ is the growth rate of particles growing into the size bin $i$, $N_{GR_{in},i}$ is the number concentration of particles able to grow into the size bin $i$ in the studied time step ($t_{step}$), which is calculated based on $GR_{in,i}$, and $\Delta D_{p,GR_{in},i}$ is the size range of those particles. Correspondingly, $GR_{out,i}$ is the growth rate of particles growing out of the size bin $i$, $N_{GR_{out},i}$ is the concentration of particles growing out of the size bin $i$ in $t_{step}$ and

$\Delta D_{p,GR_{out},i}$ is their size range. CoagS$_i$ is the coagulation sink for particles in size bin $i$, caused by larger particles, and DR$_i$ is
the loss rate of particle in the size bin $i$ due to wet and dry deposition.

For the smallest size bin ($i = 1$), the term describing the growth into the size bin is omitted, and thus the emissions calculated for the first size bin also include the flux of growing particles from below the lowest considered size. These particles can originate from primary emissions but also from atmospheric NPF. We omit the first growth term for the smallest size bin for two reasons: 1) to include the effect of atmospheric clustering on particle production and 2) because the measured
concentrations of the smallest particles, needed for calculating the flux of particles growing into the size bin, contain large uncertainties. Overall, one should note that applying Eq. (2) to determine particle number emissions includes many assumptions. In the next section, we discuss these assumptions and their validity for our data set from Beijing.

## 2.2 Main assumptions of the method

### 2.2.1 Transport

One of the main simplifications of our method is that the effect of particles advected to the measurement site is not included in Eq. (2). We assume that if we apply Eq. (2) to long enough data set and then determine the average diurnal cycle of emissions, the effect of the transport from point sources located in different directions from the measurement site is evened out. This is because the particle transport from a point source has both positive and negative contributions to particle emissions on individual days, at the moments when the wind turns to come from the direction of the source and when it turns away from
that direction. Therefore, when averaging over many days, the transport effect can be expected to become minor and the resulting emissions describe those sources that are present most of the time and distributed rather evenly in the urban region surrounding our site. For this assumption to be valid, the data set needs to be long enough, wind direction should not have a strong diurnal cycle, and the point sources should be irregularly located. If these criteria are not met, there can be some bias in the calculated particle emissions due to particle advection. In Sect. 3.5.1, we investigate this by comparing the average
emissions for different wind directions and wind speeds. Although this analysis suggests that the bias caused by particle transport is relatively minor, the source area of the emissions calculated by our method cannot be accurately determined. Furthermore, one should note that in urban environments there can be large local differences in particle emissions (Harrison, 2018), which are not captured by our method.

### 2.2.2 Mixing of boundary layer

In Eq. (2) we assume that ML is homogeneously mixed, which is not necessarily true in an urban environment, where buildings act as large roughness elements that can affect the mixing at the lower levels of boundary layer (Barlow, 2014). Studies comparing particle size distribution and aerosol chemical composition between the ground level and a height of 260 m in Beijing have shown that aerosol properties between these heights can significantly differ, depending on meteorological conditions (Du et al., 2017; Wang et al., 2018). This indicates that ML in Beijing is not always well-mixed, which may cause

us to over- or underestimate particle emissions, depending on the structure of boundary layer and the height of the particle sources.

In addition, we assume that the increase of ML in the morning causes dilution in the concentrations of all particle sizes. This is likely a good assumption for the smallest particles, which have short lifetimes and therefore are likely not present in the residual layer in the morning, when air from the residual layer is mixed with the increasing ML. However, larger particles with longer lifetime can maintain higher concentrations in the residual layer throughout the night, and thus we may overestimate the effect of dilution on their concentrations inside the ML.

### 2.2.3 Particle losses

As shown in Eqs (1) and (2), we assume that the only particle-removal mechanisms that play an important role are the coagulation scavenging by larger particles and deposition. However, it has been suggested that evaporation of traffic-originated nucleation mode particles may be significant (Harrison et al., 2016). If this is the case, we may underestimate particle number emissions, depending on how fast particles evaporate after their emission and how far the measurement site is located from the road.

In addition, when we describe the removal of particles by deposition, we assume a constant deposition rate for all particle sizes, corresponding to the lifetime of 1 week (Stocker, et al., 2013). In reality, dry and wet deposition are size- and time-dependent processes, which depend, for example, on the properties of available surfaces, boundary layer and rainfall (e.g. Laakso et al., 2003; Zhang and Wexler, 2002). Thus, a constant deposition rate can cause uncertainties in estimated emissions, especially for the largest particles for which deposition is most important due to low coagulation losses. With our assumption for the deposition rate, deposition affects significantly only the emissions of particles larger than 100 nm, by increasing their emissions by maximum of ~20% at night and less during the day.

Finally, it has been suggested that coagulation scavenging of the smallest particles may be less efficient than theoretically expected in Chinese megacities, which could explain the observed high survival probability of growing particles in NPF events (Kulmala et al., 2017). In this work, we do not consider possible ineffectiveness of coagulation scavenging, as the magnitude and size-dependence of this effect is unknown and also because we focus on days without NPF events. This may cause us to overestimate particle number emissions at the smallest ($D_p < $ ~5 nm) sizes.

### 2.2.4 Particle growth

When describing the effect of growth into and out of the size bins in Eq. (2), we assume a constant value for GR for all the size bins, although it would be possible to include the size-dependence of GR in the calculations. Zhou et al. (2020) recently showed that GR of particles between 1 and 30 nm on average increases with size at our measurement site. However, we chose to assume constant GR because of the uncertainty of the size-dependent values of GR for the whole studied size range, and to simplify the interpretation of the results. With a constant GR, the terms in Eq. (2) describing the growth into and out of the

size bin offset each other if particle concentration does not significantly change with size. The sensitivity of the results to GR and its size-dependency is discussed in Sect. 3.5.2.

### 2.2.5 Coagulation source

In Eq. (2) we do not consider the production of particles into size bin $i$ due to the collision between two smaller particles resulting in a particle in size bin $i$. The error caused by this simplification can be estimated to be minor, because coagulation coefficients are highest for the particles with a large size difference and their collisions have only little effect on the size of the larger particle. Cai et al. (2018) applied a population balance method to study how transport affects temporal evolution of particle size distribution on an NPF event day in Beijing, and found that the source of particles due to coagulation of smaller particles was negligible compared to the coagulation losses of the particles.

### 2.3 Application of the method to measurements in Beijing

We applied the introduced method to estimate particle number emissions in Beijing, China, using measurements performed at the measurement station of Beijing University of Chemical Technology (BUCT) during January 2018 – March 2019. The station is located in the western part of Beijing (39$^{\circ}$ 56' 31" N, 116$^{\circ}$ 17' 50" E), about 150 m south-east from the closest busy road and 550 m west from the 3rd Ring Road of Beijing. The location of the measurement site is shown in Fig. 1 with respect to urban Beijing and its surroundings. The urban region with high population density (Fig. 1b) and high emissions of $PM_{2.5}$ and different trace gases ($NO_x$, CO and $SO_2$) based on emission inventories (Fig. A1) extends ~20 km west, ~100–200 km east, and ~50 km north and south of our site.

For particle size distribution data, we used data measured with a Diethylene Glycol Scanning Mobility Particle Sizer (DEG-SMPS; Cai et al., 2017; Fu et al., 2019; Jiang et al., 2011) and a custom-made Particle Size Distribution (PSD; Liu et al., 2016) system. The DEG-SMPS measures particle sizes between 1 and 6.5 nm (electrical mobility diameter) and the PSD system particle sizes between 3 nm and 10 μm, using a combination of a homemade Nano-SMPS (3–55 nm, electrical mobility diameter), a homemade Long-SMPS (25–650 nm, electric mobility diameter), and a TSI 3321 aerodynamic particle sizer (0.55–10 μm, aerodynamic diameter). We corrected particle diffusion losses, bipolar charging efficiency, multiple charging, and detection efficiency, when inverting the size distribution data. To obtain the final size distribution for the size ranges where different instruments overlap, we calculated the weighted average of size distributions measured with different instruments. The days when the whole particle size distribution was not measured reliably due to instrument malfunctioning were disregarded. The final corrected data set includes 136 days of particle size distributions between 1 nm and 10 μm, covering months from October to May. For more details of the particle size distribution measurements performed at the BUCT station, see Zhou et al. (2020).

Based on the particle size distribution data, we classified the days into days with an NPF event and days without an event. A day was classified as an NPF event day if an appearance of a new mode of sub-10 nm particles and the further growth of this mode was observed, and it was not clearly linked to particle emissions from traffic.

MLH was obtained from ceilometer measurements (CL-51; Vaisala Inc, Finland) of the optical backscattering by applying a three-step idealized-profile (Eresmaa et al., 2012). Because ceilometer data were not available for every day with particle size

distribution data, we calculated the average diurnal cycles of MLH for NPF event days and nonevent days and used them when applying Eq. (2). This is justified as we study the average diurnal cycle of particle number emissions, instead of their day-to-day variation.

For GR we used a constant value of 3 nm/h for all the size bins, which corresponds to typical GR between 3 and 7 nm at the station during the measurement period (Zhou et al., 2020). To describe the losses of particles by coagulation scavenging, we

calculated CoagS for each size bin $i$ from the particle size distribution data, based on the coagulation coefficients between particles in size bin $i$ and larger particles (Kulmala et al., 2001).

When applying Eq. (1) to our data set, we calculated particle number emissions to 22 particle size bins with the lower limit $D_p$ and the upper limit $D_p \times 4/3$, between 2.0 nm and 1.1 μm. After calculating particle number emissions for each day, we determined the average diurnal cycle of particle number emission size distributions separately on NPF event days and non-

230 event days.

We compared the emissions determined with our method to those calculated with the GAINS model (Paasonen et al., 2016). The GAINS emissions were retrieved from the model web page (https://www.iiasa.ac.at/web/home/research/researchPrograms/air/PN.html, providing calculated emissions for years 2010, 2020 and 2030) for the grid cell of 0.5° x 0.5°, in which the center of Beijing is located. We used emissions calculated for the

235 year 2010 based on the results of Paasonen et al. (2016) that indicate that the emissions for the year 2010 have less uncertainties associated to them than the corresponding values for the year 2020. In addition, to gain insight into the effects of particle transport and the source area of our method, we utilized emissions of $PM_{2.5}$, $NO_x$, CO and $SO_2$ obtained from the MIX emission inventory (Li et al., 2017), which is the combined result of the best available regional scale emission inventories in Asia. The MIX inventory used here describes emissions for the year 2010 on a 0.25° × 0.25° grid and the data is available online

(http://www.meicmodel.org/dataset-mix.html). In this study, the emissions of different trace gases are used to describe the general activity levels of different kinds of combustion sources, which also emit particles.

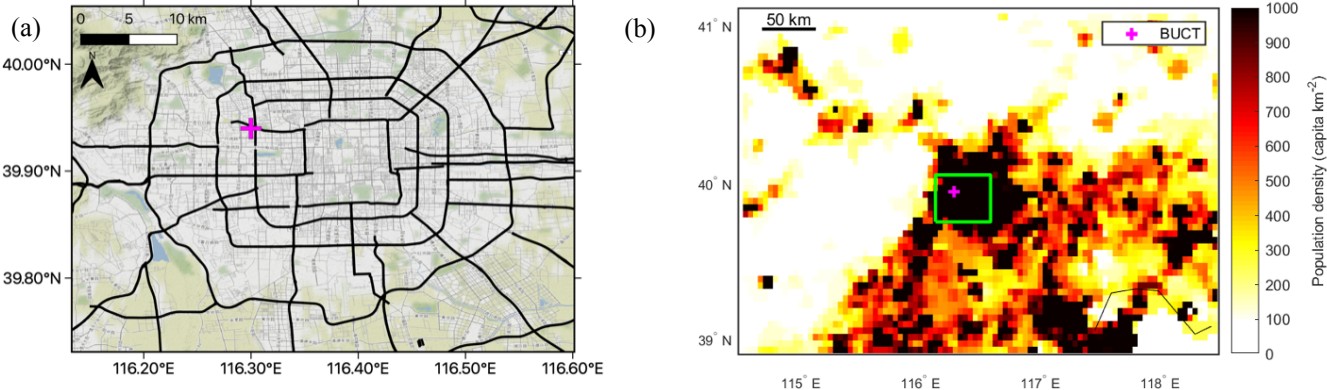

Figure 1. The maps of (a) urban Beijing and its main roads, and (b) the region around Beijing with the population density (year 2015) shown as color. The location of the measurement site of BUCT is shown with a magenta cross in both maps. The green rectangle in (b) corresponds to the region shown in (a). In (a) the map data obtained from Stamen Design (CC BY 3.0) and © OpenStreetMap contributors 2020. Distributed under a Creative Commons BY-SA License (ODbL). In (b) the population density data obtained from Gridded Population of the World (GPWv4.10; CC BY 4.0).

## 3 Results and discussion

### 3.1 Diurnal cycles of MLH and particle number size distributions

During the measurement period, 44% of the days were classified as NPF event days. Figure 2 presents the average diurnal cycle of MLH and its time derivative ($d$MLH/$d$t) on NPF event days and nonevent days. Both on NPF event days and nonevent days, MLH starts to increase after 6:00 in the morning and reaches its maximum around 15:00. However, on NPF event days MLH reaches clearly higher values (the maximum height ~2200 m) than on nonevents days (the maximum height ~820 m), and thus the time derivative of MLH is larger on NPF event days. Note that the time derivative is shown only for the mornings, when MLH increases, causing dilution of particle concentrations.

The average diurnal variation of particle number size distribution on NPF event days and nonevent days is shown in Fig. 3. On nonevent days particle concentrations between ~6 and 150 nm exhibit clear maxima during morning (06:00–12:00) and evening (17:00–23:00) hours. This is caused by emissions of particles from traffic and possible other sources, and the growth of the emitted particles. On NPF event days, primary particle emissions can also be observed, but the time-evolution of the particle size distribution is dominated by the appearance of a high number of sub-5 nm particles between about 08:00 and 17:00 and their growth to larger sizes. One should note, though, that the growth of all sub-5 nm particles, especially those

appearing in the afternoon, cannot be observed at the measurement site. This causes difficulties when estimating particle number emissions for NPF event days, as discussed in the next section.

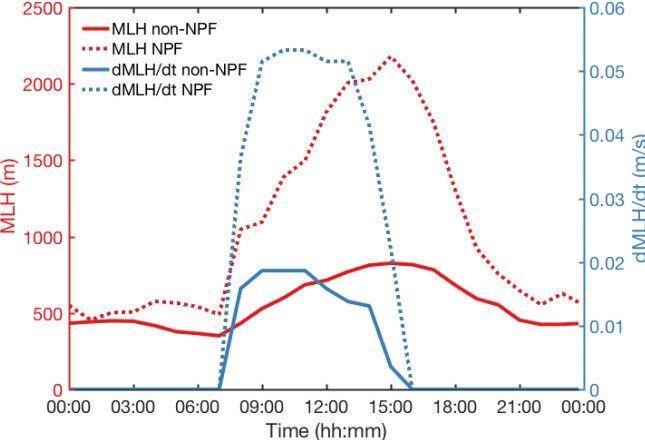

**Figure 2: Average diurnal variations of MLH (mixing layer height; red lines and left y-axis) and the time derivate of MLH when it is positive (blue lines and right y-axis) on days without NPF events (solid lines) and on NPF event days (dashed lines).**

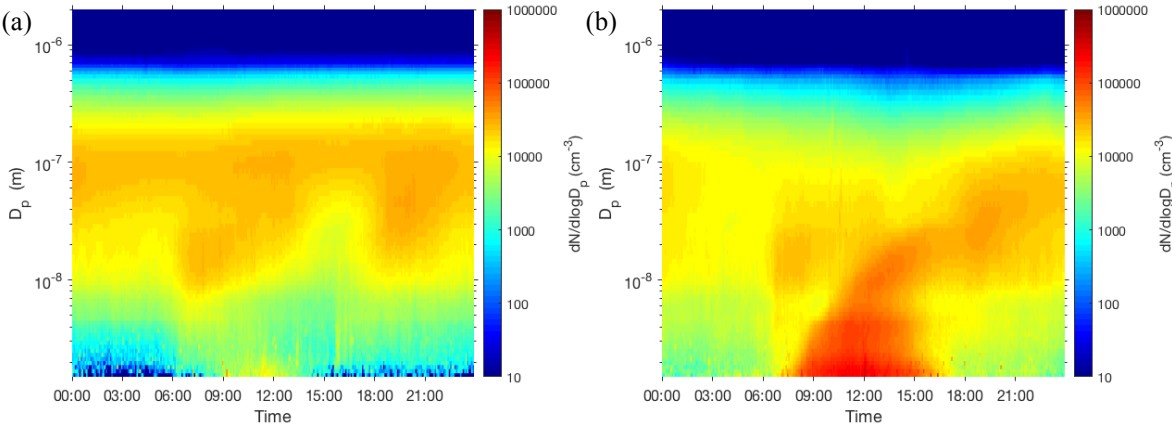

**Figure 3: Average diurnal variation of particle number concentration size distributions (a) on days without NPF events and (b) on NPF event days.**

### 3.2 Diurnal cycles of particle number emissions

We calculated the average diurnal cycle of particle number emission size distributions separately for NPF event days and nonevent days (Fig. 4). On nonevent days the time-evolution of particle number emissions looks reasonable. The emissions at

275 almost all studied sizes are highest during morning (06:00–12:00) and evening (17:00–22:00), which probably is, at least

partly, linked to particle emissions from traffic. The connection to different sources and the differences in particle emissions between different sizes are discussed in more detail in the next sections.

On NPF event days, the time-evolution of particle number emission size distributions looks less plausible. A strong production of sub-3 nm particles by atmospheric NPF can be observed during the day, as expected. However, on NPF event days we also see a clearly higher production of particles larger than 3 nm (~3–5 nm and ~7–20 nm) than on non-event days, simultaneously or immediately after particles are produced to the smallest size bin (~2–3 nm). This indicates that our calculations are unable to accurately describe particle dynamics in NPF events, and therefore the contribution of NPF can also be observed at sizes larger than 3 nm. There can be several reasons for this. For example, higher particle formation rate at the higher levels of the boundary layer could lead to an increasing particle concentration with increasing diameter, when more numerous particles from above would be transported to the measurement site and detected after their initial growth during the transportation. In addition, the results can be affected by time- and size-dependent variation in particle GR (see Sect. 3.5). Other possible reason is measurement uncertainties, which can be expected to be highest at the smallest sizes and around the sizes where the particle size distribution instrument changes (see Sect. 2.3). The calculated particle emissions for NPF event days look unreliable also because of the distinct minimum visible between 5.5 and 7.2 nm. The minimum is likely mainly caused by not all sub-6 nm particles growing to larger sizes, as discussed in Sect 3.1. Therefore, when we subtract the term describing the growth into the bin of 5.5–7.2 nm (see Eq. 2), we end up with too small, even negative emissions. In addition, the change of the instrument around that size range may also affect the calculated emissions. Finally, the differences in calculated emissions on NPF event days and non-event days can also be partly due to differences in prevailing wind direction on event and non-event days (see Sect. 3.5). Overall, due the difficulties in describing particle dynamics on NPF event days, we focus on determining particle number emissions on nonevent days. Determining the exact contributions of primary particle emissions and NPF to particle number concentrations on NPF event days requires further work and it will be a subject of future study.

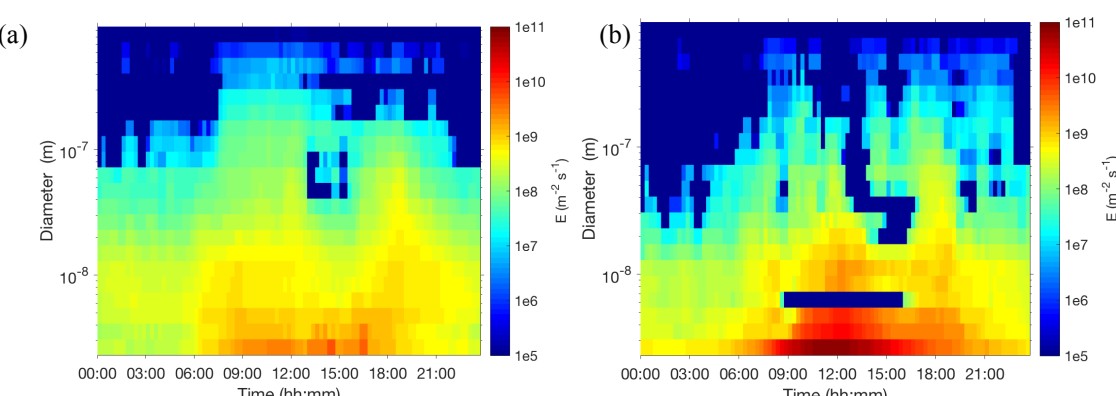

**Figure 4: Average diurnal variation of particle number emission size distributions (a) on days without NPF events and (b) on NPF event days.**

**3.3 Connection between variation of particle number emissions and traffic**

To investigate the variation of particle number emissions in more detail, we determined the diurnal cycle of particle number emissions for different size ranges (Fig. 5a in a linear scale and Fig. A2 in a logarithmic scale). We also studied the diurnal cycle of boundary layer burden of nitrogen oxides ($NO_x$), which is calculated as the product of $NO_x$ concentration and MLH and which roughly represents the diurnal variation of $NO_x$ emissions. As shown by Fig. 5b, the estimated $NO_x$ emissions have a maximum around 09:00, linked to morning traffic, while they do not have a clear afternoon or evening maximum, likely due to fast photochemical loss of $NO_x$ (Lu et al., 2019). Cai et al. (2020) used EMBEV-Link (Link-level Emission factor Model for the BEijing Vehicle fleet; Yang et al., 2019) model to estimate the diurnal cycle of $PM_{2.5}$ emissions at our measurement site. According to the modeling results, $PM_{2.5}$ emissions originating from gasoline vehicles in urban Beijing start to increase before 06:00 in the morning, reach the first maximum around 7:00–8:00 and the second maximum around 17:00–18:00, after which they decrease to lower night-time values. However, the modelled $PM_{2.5}$ emissions from diesel vehicles are highest at night (Cai et al., 2020).

Figure 5a shows that the particle emissions to the smallest studied size bin (~2–3 nm) (which also include the growth of the particles from smaller sizes) increase in the morning, reach a first maximum just before noon, and show two other peaks around 14:00 and 16:00. The noon-time maximum, which is also observed on NPF event days (figure not shown), suggests that formation of sub-3 nm particles by clustering of vapor molecules can take place on nonevent days, but because the growth of particles to larger sizes is not seen, it is not defined as an NPF event. Weak production of sub-3 nm particles can also be observed in the average diurnal cycle of particle number concentrations on non-NPF event days (Fig. 3). In addition to atmospheric clustering, it is possible that some of the sub-3 nm particles originate from traffic (Rönkkö et al., 2017).

The emissions to the size range between 3 and 6 nm are highest between 08:00 and 12:00 and around 14:00 and 17:00 (Fig. 5a). The morning maximum coincides with the morning maximum of estimated $NO_x$ emissions (Fig. 5b), suggesting that traffic contributes to particle emissions into this size range. The importance of traffic emissions is also supported by the fact that the diurnal cycle of emissions is roughly similar to the diurnal cycle of modelled $PM_{2.5}$ emissions from gasoline vehicles in Cai et al. (2020), which have maxima around 7:00–8:00 and 17:00. In addition, clustering of atmospheric vapors and the following growth to 3–6 nm sizes can contribute to the emissions calculated to this size range, as atmospheric clustering seems to occur also on nonevent days. This is further supported by our analysis in Sect. 3.5.

The emissions to the size ranges of 6–30 nm and 30–100 nm have quite similar diurnal cycles with the first maximum between 08:00 and 12:00 and the second, slightly higher maximum after 18:00 (Fig. 5a). The morning maxima indicate particle emissions from traffic to these size ranges too. The fact that the evening maxima are higher than the morning maxima suggest either higher emissions from traffic to these size ranges at this time of the day, or then possible contribution from other emission sources (see the discussion in the next section).

The emissions to the largest size range (100–1000 nm) are overall low, exhibiting one clear maximum around 10:00 and another, much less pronounced one, around 18:00 (see Fig. A2). Although the morning maximum could be related to emissions

from traffic, the fact that it is much more distinct than the evening maximum suggests that it may be partly caused by overestimating the effect of dilution due to increase of MLH in the morning. As discussed in Sect. 2.2, it is unlikely that the concentrations of particles larger than 100 nm always decrease with increasing MLH, as assumed in Eq. (2).

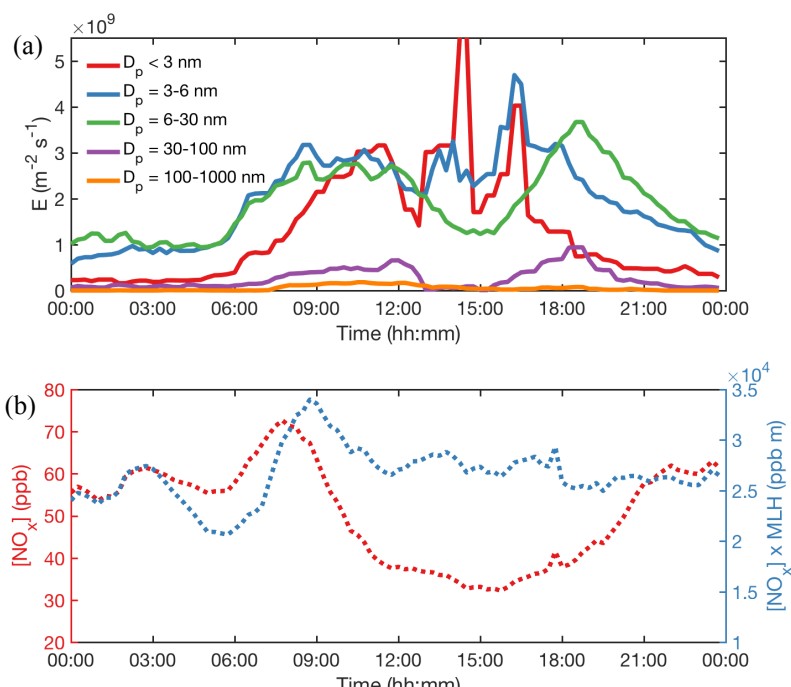

**Figure 5: Average diurnal cycles of (a) particle number emissions into different size ranges on non-NPF event days, (b) the concentration of $NO_x$ (nitrogen oxidizes) and its product with MLH (mixing layer height). For particle number emissions depicted in a logarithmic scale, see Fig. A2 in Appendix.**

### 3.4 Average size distributions of particle number emissions

To get more insight into particle emissions at different sizes, we studied the average particle number emission size distributions at different times of the day: early morning (06:00–08:00), late morning (09:00–11:00), evening (18:00–20:00) and midnight (00:00–02:00) (Fig. 6; see also Fig. A3). Clear differences between the size distributions at different hours can be observed, indicating the production of particles from different sources.

Strong production of the smallest ($D_p$ < 3 nm) particles is observed at 09:00–11.00 (Fig. 6), which is likely connected to atmospheric cluster formation, as discussed above. The production of this sized particles is moderate also in the early morning and evening, and non-negligible even at night. Recently, atmospheric NPF in Beijing was suggested to start with clustering between sulfuric acid and an amine (Deng et al., 2020) and thus this is likely the main mechanism for the observed formation of sub-3 nm particles. This mechanism is stronger during the day, due to photochemical production of sulfuric acid, but it is possible that these clusters also form at night-time. On the other hand, traffic emissions may also contribute to the production

of sub-3 nm particles, as dilution and cooling of traffic exhaust has been shown to produce a high number of sub-3 nm particles (Rönkkö et al., 2017).

The size distributions of particle number emissions show a maximum around 10 nm at all times (Fig. 6). The diurnal cycle of emissions into this size range (Figs 4 and 5) indicate that this maximum is likely caused by traffic emissions. This is supported by laboratory measurements showing that traffic exhaust contains nucleation mode particles (Rönkkö et al., 2007; Shi and Harrison, 1999), which in some conditions have a mode diameter of ~10 nm (Rönkkö et al., 2017). In addition, in road-side measurements of 1–1000 nm particle number concentrations, particle modes around 1–3 nm and 10 nm have been observed in urban and semi-urban background conditions (Hietikko et al., 2018; Rönkkö et al., 2017).

At sizes between ~15 and 50 nm, the emissions are clearly highest at 18:00–20:00 (Fig. 6). Although traffic likely contributes to emissions into this size range, high emissions in the evening can indicate the contribution of some other source, such as cooking activities. The contribution of cooking emissions at this time is supported by studies applying PMF analysis to chemical composition and particle size distribution data from Beijing, which have found cooking-related factors peaking around 19:00–20:00 (Cai et al., 2020; Hu et al., 2017; Liu et al., 2017). In a study by Cai et al. (2020), the cooking-related particle number size distribution factor had a GMD of ~50 nm. In studies focusing on cooking emissions, Chinese cooking has been found to typically produce particles with the mode diameter ranging from 20 to 100 nm (Zhao and Zhao, 2018).

There is a weak maximum visible in the particle size distribution around 100 nm at 09:00–11:00, also seen as a separate shoulder in the logarithmic emission size distribution at 6:00–8:00 (Fig. A3). As discussed above, this maximum may be related to traffic but can also be due to overestimation of the dilution effect for larger particles. Generally, the emissions at sizes larger than 100 nm are low, and particle number emissions around our measurement site seem to be dominated by emissions of smaller particles, especially those in nucleation mode ($D_p < 30$ nm).

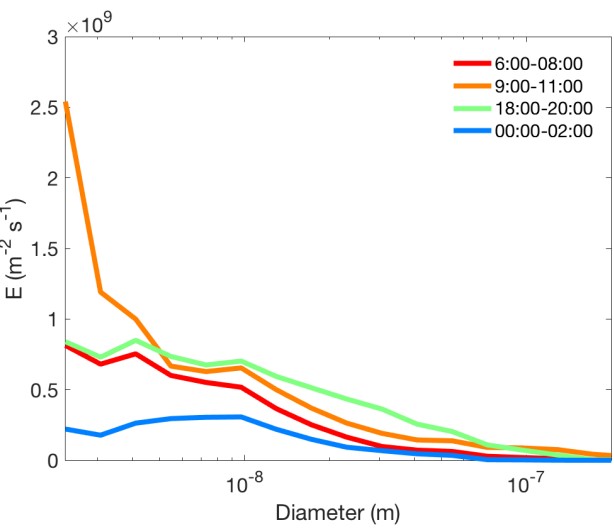

**Figure 6: Average particle number emission size distributions on non-NPF event days at different times. For the size distributions depicted in a logarithmic scale, see Fig. A3 in Appendix.**

## 3.5 Sensitivity of the calculated emissions to wind conditions and particle growth rate

### 3.5.1 Sensitivity to wind direction and wind speed

To investigate how our results are influenced by transport of particles from sources at different directions and distances from our site, we studied how wind direction and wind speed affect the calculated particle emissions. First, we investigated the frequency of different wind directions during daytime (09:00–15:00) and at night (21:00–03:00) on non-NPF event days and found that north-western winds are most frequent during daytime and south-eastern winds at night (Fig. A4). Then, we selected the nonevent days with predominantly south-eastern winds (wind direction from the sector 45°–225° for more than 95% of the time; 18 days) and with predominantly north-western winds (wind direction from the sector 225°–45° for more than 95% of the time; 26 days), and determined the average particle number emission size distributions for these days. One should note that because of the limited number of days for these two cases, the average emission size distributions are sensitive to sudden changes in particle concentrations on those days.

As shown in Fig. 7a, there are apparent differences in the emission size distributions between the studied wind directions (see also Fig. A5a). First of all, when wind is coming from the north-western directions, the production of the smallest particles is stronger. This is clear especially at 09:00–11.00, suggesting that the difference is caused by northern winds favoring atmospheric cluster formation. It is known that in Beijing NPF events typically start when wind is bringing relatively clean air from the northern directions (Wehner et al., 2008). At 09:00–11.00 the higher particle production linked to north-western winds can be seen up to ~6 nm, which indicates that cluster formation and the following growth can contribute to the calculated emissions up to 6 nm sizes even on non-NPF event days. In addition to particle formation, the stronger production of the smallest particles linked to north-western winds could be due to their higher emissions to the north-west of the measurement site.

The second clear difference in the emission size distributions between the wind directions is higher emissions of particles larger than 7 nm in the morning and at night when wind is coming from the south-east (Fig. 7a and Fig. A5a). At 06:00–08:00, the emissions for particles between 7 and 100 nm are higher by a factor of ~1.4–2 for south-eastern directions. Thus, there seems to be inhomogeneities in particle number emissions around our measurement site, with stronger emissions in the south-eastern directions in the morning, or, as discussed below, with a further extending high emission region in that direction. However, at 18:00–20:00 the emissions for particles between 10 and 50 nm are higher with north-western winds, by up to a factor of ~1.6, suggesting higher emissions in that direction. Still, the differences between the emissions with different wind directions are relatively minor when considering all the assumptions behind our method (see Sect. 2.2). When looking at the population density in the region surrounding our measurement site (Fig. 1b) and the emissions of $PM_{2.5}$ and trace gases based on emission inventories (Fig. A1), a strong decline in particle emissions can be expected ~20 km west and ~50 km north of our site, and a moderate, more gradual, decline ~100–200 km east and ~50 km south of the site. Thus, the difference of up to

a factor of 2 between north-western and south-eastern directions in our results indicates that most of the emissions obtained with our method originate within a radius of a few tens of km from our site, inside urban Beijing. However, one should note that there are two busy roads located close to our measurement site, which likely enhances the calculated emissions relative to the average emissions of the urban region.

We also investigated the effect of wind speed on the calculated emissions. We did this by determining the average particle number emission size distributions for days when 1-hour averaged wind speed was predominantly over 1.1 m/s (20 days) and for days when averaged wind speed was predominantly below 0.6 m/s (10 days). Figure 7b shows that the differences in the emissions between different wind speeds are generally minor (see also Fig. A5b). During the day, the ratio between the emissions at low and high wind speeds varies mostly between 0.6 and 1.3 at different sizes. In the evening, the emissions for the smallest particles are higher at higher wind speeds, which is likely connected to atmospheric cluster formation. However, at the same time the emissions for particles between 10 and 100 nm are higher at lower wind speeds, by up to a factor of ~2.5. Higher particle emissions at lower wind speeds are expected as then particles have more time to accumulate in the air mass traveling to our site over the urban region. The reason that this is clearest in the evening may be a more stable boundary layer at that time of the day. Still, the fact that the differences in the emissions between different wind speeds are rather small, supports the idea that the emissions calculated with our method are mainly affected by particle sources within urban Beijing. This is also indicated by generally low emissions of particles larger than 100 nm, for which the effect of transport from sources outside the urban region should be most important, due to their long lifetime. Determining more quantitatively the impact of particle transport on the calculated emissions would require modelling of the transport of particles from different sources to our site under different meteorological conditions, which is outside the scope of this study. For this reason, the emissions calculated with the current version of our method should not be considered precise.

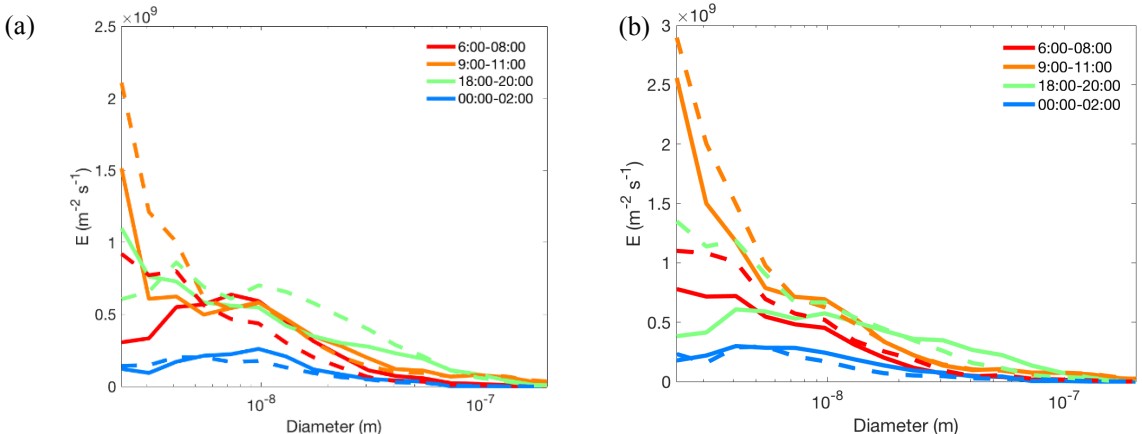

**Figure 7: Average particle number emission size distributions for non-NPF event days (a) when wind is coming from the south-eastern directions (45°–225°; solid lines) and from the north-western directions (225°–45°; dashed lines), and (b) when wind speed**

**is predominantly below 0.6 m/s (solid lines) and over 1.1 m/s (dashed lines). For the size distributions depicted in a logarithmic scale, see Fig. A5 in Appendix.**

### 3.5.2 Sensitivity to particle growth rate

To study the sensitivity of our results to size-dependency of particle GR, we determined particle number emissions by assuming that GR increases with increasing particle diameter. We utilized the medians of particle GRs observed at the site for three size ranges (< 3 nm, 3–7 nm and 7–25 nm) (Zhou et al., 2020) and determined GR for each size bin in our emission calculations based on a fit to (GR, log($D_p$)) data (Fig. A6). As shown by Fig. 8a and Fig. A7a, at sizes below ~20 nm emissions calculated with the increasing GR are very close to the emissions calculated with the constant value that we assume in this study (GR = 3 nm/h). At larger sizes, where GR estimated from the fit becomes high, emissions calculated with increasing GR become mostly smaller than emissions calculated with GR = 3 nm/h (Fig. A7a).

To get more insight into the effect of the value of GR on calculated emissions, we determined particle number emissions with two times higher and lower GR than our normal assumption. Figure 8b shows the average size distributions of particle number emissions when assuming GR = 1.5 nm/h and GR = 6 nm/h (see also Fig. A7b). Generally, the particle number emission size distributions are quite similar in the two cases, except at the smallest sizes. At 09:00 and 11:00, the emissions to the smallest size bin are by a factor of 2 higher with GR = 6 nm/h, which results from the fact that when applying Eq. (2) to the smallest bin, the term describing growth into the bin is omitted (see Sect. 2.1). In addition, between ~3 and 4 nm, there is a minimum in the emission size distribution with GR = 6 nm/h. This is caused by emissions into this size bin becoming negative around midday (figure not shown), which indicates a too high value of GR. The negative emissions are due to strongly decreasing particle concentration with diameter in that size region, which causes the term describing the growth into the size bin in Eq. (2) to be clearly higher than the term describing the growth out of the bin. At larger sizes and at other times of the day, the differences in the emission size distribution with different GRs are subtler. If the particle concentration decreases with increasing particle diameter in the studied size range, emissions become lower with higher GR, and if particle concentration increases with increasing diameter, the opposite is true. Overall, we can conclude that the calculated particle emissions are sensitive to the value of GR only at the smallest sizes, where particle number concentration changes steeply with size. At these sizes, GR = 3 nm/h is a good estimate for our measurement site based on the results by Zhou et al. (2020).

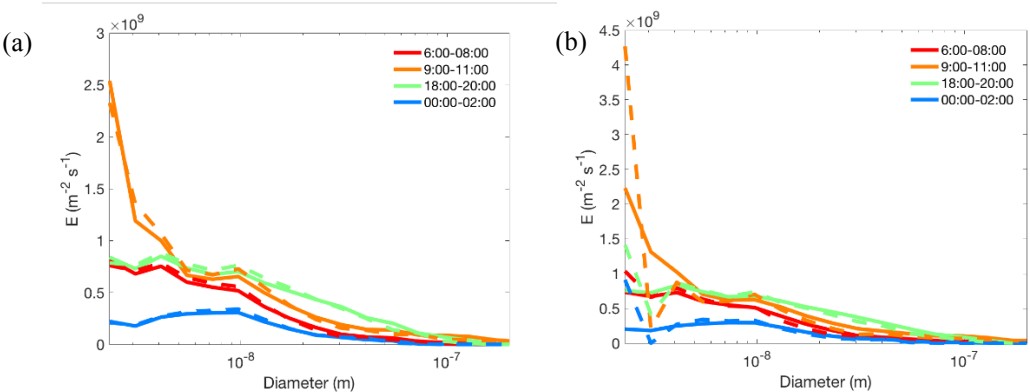

**Figure 8: Average particle number emission size distributions for non-NPF event days assuming (a) GR = 3 nm/h (solid lines) and GR that increases with size (dashed lines, see text for details), (b) GR = 1.5 nm/h (solid lines) and GR = 6 nm/h (dashed lines). For the size distributions depicted in a logarithmic scale, see Fig. A7 in Appendix.**

## 3.6 Comparison with particle number emissions from GAINS model

We compared our results to annual particle number emissions determined for approx. $50 \times 50$ km$^2$ grid cell around downtown

Beijing with the GAINS model. This was done by calculating the annual sum of the emissions to different size bins, based on particle number emissions determined for non-NPF event days. It should be noted, though, that we used the GAINS emissions calculated for the year 2010 and the number emissions have likely changed since then. Figure 9 shows that the annual particle number emission size distributions obtained with the two methods are clearly different (see also Fig. A8). In the GAINS model, the particle emissions have a unimodal distribution with a peak at ~50 nm, while our

calculated annual emissions show multiple peaks and clearly higher particle emissions below 60 nm than GAINS (note that the smallest size bin in GAINS is 3–10 nm). However, at sizes above 60 nm, the two methods agree remarkably well. The large grid size in GAINS partly explains the lower emissions below 60 nm. Our measurement site is located close to two busy roads, and thus the contribution of traffic emissions to the observed emission size distribution can be expected to be higher than to the more regional scale emissions obtained from GAINS. Paasonen et al. (2016) also suggested that the emissions

of particles with diameters below 30 nm are underestimated in GAINS, because the experimentally determined emission factors for many sources include only particles that are nonvolatile (after heating) and/or particles larger than 10 nm in diameter.

When calculating the total annual particle number emissions to the sizes between 3 and 1000 nm, our method gives clearly higher particle number emissions ($1.1 \times 10^{17}$ m$^{-2}$) than GAINS ($1.4 \times 10^{16}$ m$^{-2}$). Although the values of particle number emissions

determined with our method should not be considered exact, due to the assumptions of the method and contribution of atmospheric cluster formation (see Sects 2.2 and 3.5), the vast difference between our calculations and GAINS model highlights the need for increased understanding of anthropogenic particle number emissions, especially for sizes smaller than

60 nm. However, the similarity of the emissions at sizes above 60 nm from GAINS and our method gives confidence in the ability of the both methods to yield reasonable estimates for particle number emissions. It also suggests that the emissions obtained with our method originate from the area approximately of the same size as the chosen grid size of GAINS, i.e. 50 × 50 km$^2$, which is consistent with our estimation in Sect. 3.5.1.

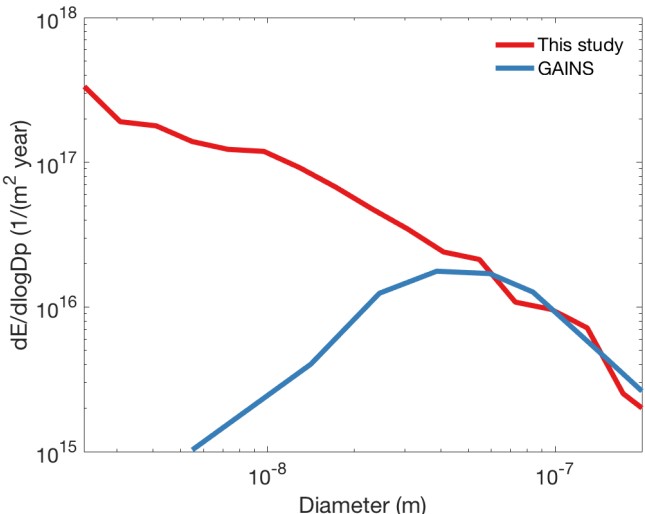

**Figure 9: Annual sum of particle number emissions at different sizes (normalized with the width of each size bin) based on particle number emissions calculated for non-NPF event days in this study (red line) and the GAINS model (blue line). In this study, the emissions to the smallest sizes include contribution from atmospheric clustering, which is not considered in the GAINS model. For the size distributions depicted in a linear scale, see Fig. A8 in Appendix.**

## 4 Conclusions

Currently, there is a lack of knowledge of size distributions of atmospheric particles emitted from anthropogenic sources. In this study, we developed a novel method for determining size-resolved particle number emissions, using measured particle size distributions. Our method is based on solving particle number emissions to different size bins from a balance equation, which considers the changes in the particle number concentration due to the direct emissions, growth into and out of the size bin, losses due to coagulation and deposition, and the dilution linked to increase of MLH. We applied this method to determine the average particle number emission size distribution and its diurnal cycle in Beijing, China. Because we found that our method cannot accurately describe the particle dynamics on NPF event days, we focused on studying emissions on days without NPF events.

We observed strong production of the smallest ($D_p < 6$ nm) particles from morning to noon, likely resulting from the formation of nanometer-sized particles by clustering of atmospheric vapors, which can occur also on non-NPF event days. We found that particle number emissions to the sizes between 6 and 100 nm are highest during morning and evening rush hours, indicating

that traffic is the major source of the emissions into this size range. This is also supported by our finding that the emission size distribution has a peak around 10 nm, consistently with earlier observations on traffic-originated particles. In addition, other sources, such as cooking activities, may also contribute to particle number emissions, particularly in the evening at sizes between 15 and 50 nm. The emissions to 100–1000 nm size range were found to be low. In general, the average contributions of different size ranges to the calculated total annual emissions are 24% for $D_p < 3$ nm, 36% for $D_p = 3–6$ nm, 34% for $D_p = 6–30$ nm, 5% for $D_p = 30–100$ nm, and 1% for $D_p = 100–1000$ nm. Thus, our results suggest that particle number emissions around our measurement site are dominated by emissions of nucleation mode ($D_p < 30$ nm) particles.

To assess the effect of particle transport on the calculated emissions, we investigated the sensitivity of the emission size distributions to wind conditions. We found that there are differences in calculated particle number emissions between different wind directions, likely resulting from differences in the strength of atmospheric clustering and particle emissions, and in the extent of the region with high emissions in different directions. The calculated emissions also slightly depend on wind speed. However, the differences between different wind directions and wind speeds are relatively minor, which indicates that the emissions obtained with our method mainly originate within the radius of a few tens of km from our site. We also studied the effect of particle GR on calculated emissions and found that the emissions are sensitive to GR only at the smallest sizes, where particle concentration changes steeply with size.

We compared our results to annual particle number emissions determined for Beijing with the GAINS model. The emissions of particles smaller than 60 nm determined with GAINS are significantly lower than our calculated emissions. However, at sizes above 60 nm our method and GAINS agree very well, giving confidence in their ability to estimate particle number emissions. Part of the difference in emissions of below 60 nm particles can be explained by the fact that the emissions calculated with our method can be affected by atmospheric cluster formation and proximity of two busy roads. The vast difference still indicates that the emissions of the smallest particles in GAINS are severely underestimated and that it is crucial to improve their description.

Overall, our method was found to produce the size distribution of particle number emissions and its diurnal variation in Beijing in a plausible way. Further work is still needed to be able to determine the contributions of particle number emissions and NPF to particle concentrations on NPF event days. To improve the method, more knowledge of particle dynamics in urban environments is needed, such as the loss rates of different sized particles due to evaporation and deposition and the impacts of the urban boundary layer development on particle dynamics. Further work is also required to quantify the effect of particle advection on the calculated emissions by modelling the transport of particles from different sources. In the future, our method can be used to provide new knowledge of particle number emissions in different environments. This is needed for validating and improving modelled particle emissions, which are essential when making decisions on future air quality strategies.


**Appendix A**

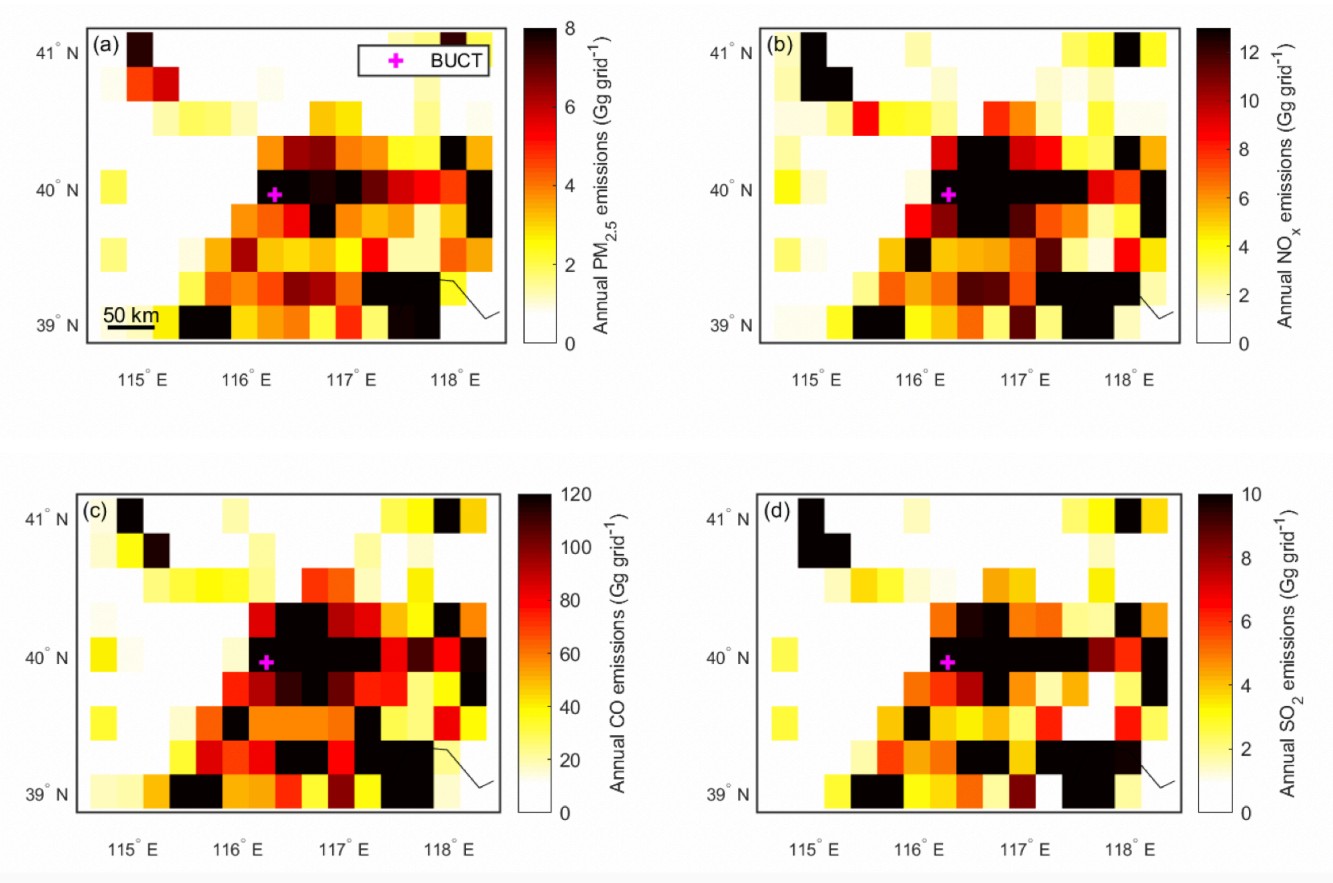


**Figure A1. Annual emissions of (a) PM₂.₅, (b) NOₓ, (c) CO and (d) SO₂ for the year 2010 based on the MIX emission inventory (Li et al., 2017) in the region around Beijing.**


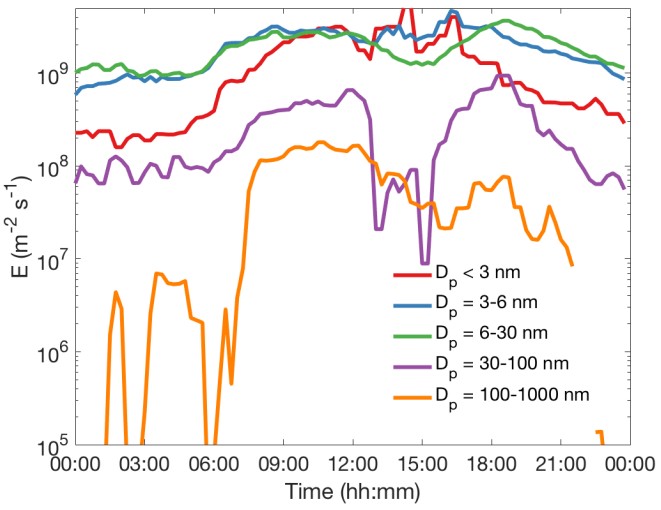

**Figure A2. Average diurnal cycles of particle number emissions into different size ranges on non-NPF event days.**

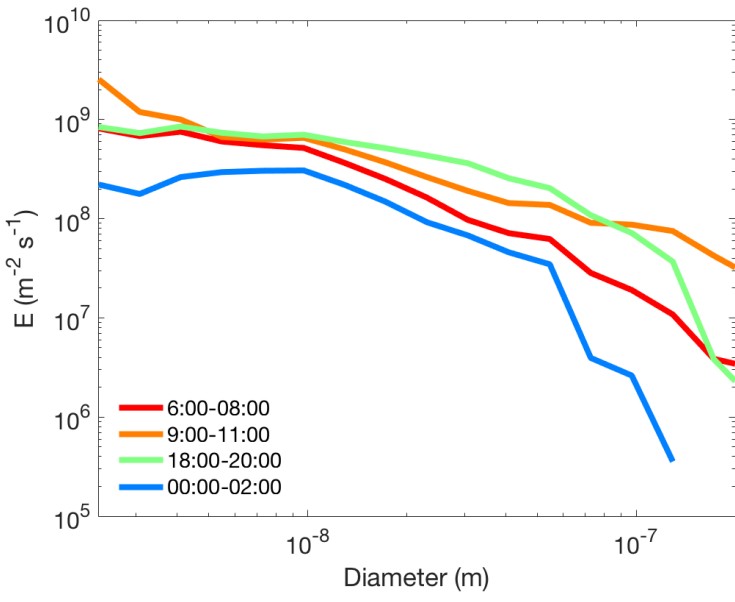

**Figure A3. Average particle number emission size distributions on non-NPF event days at different times.**

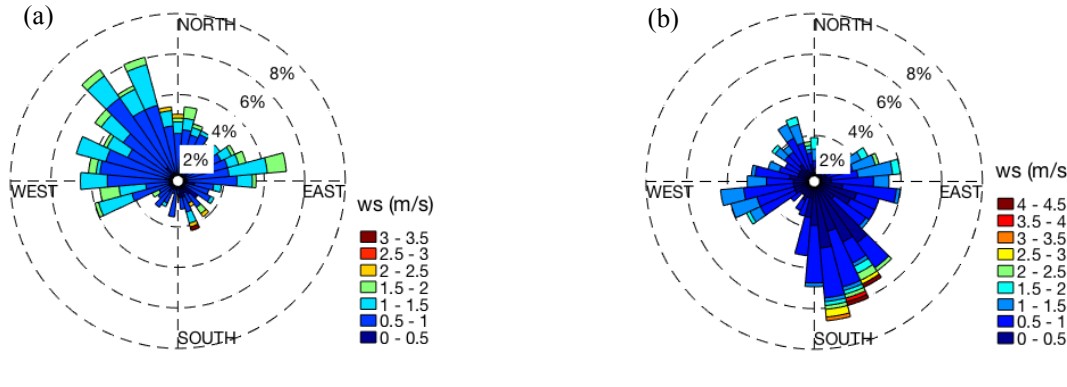

**Figure A4. Wind roses for (a) daytime (09:00–15:00) and (b) night-time (21:00–03:00) for non-NPF event days. The lengths of the wedges show the frequency of each wind direction and the colors illustrate the frequency of different wind speed values (ws).**


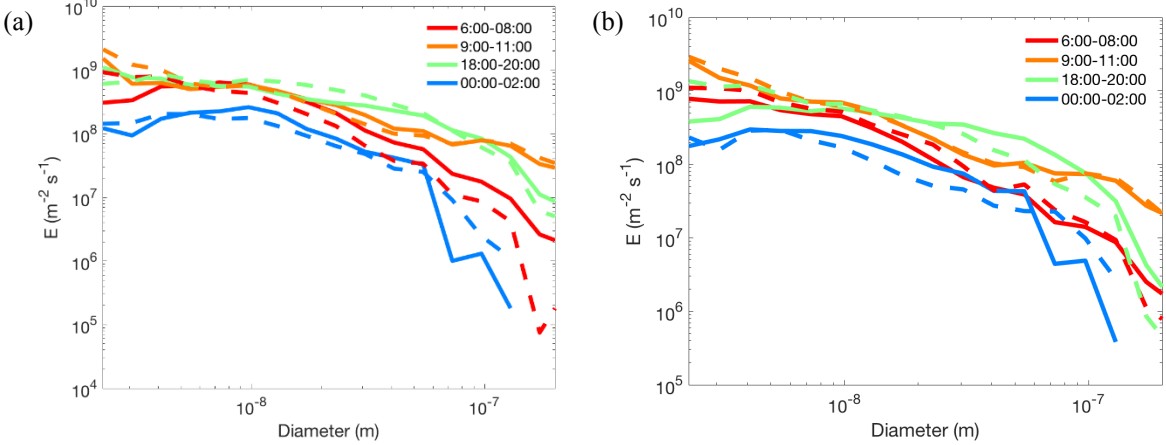

**Figure A5: Average particle number emission size distributions for non-NPF event days (a) when wind is coming from the south-eastern directions (45°–225°; solid lines) and from the north-western directions (225°–45°; dashed lines), and (b) when wind speed is predominantly below 0.6 m/s (solid lines) and over 1.1 m/s (dashed lines).**


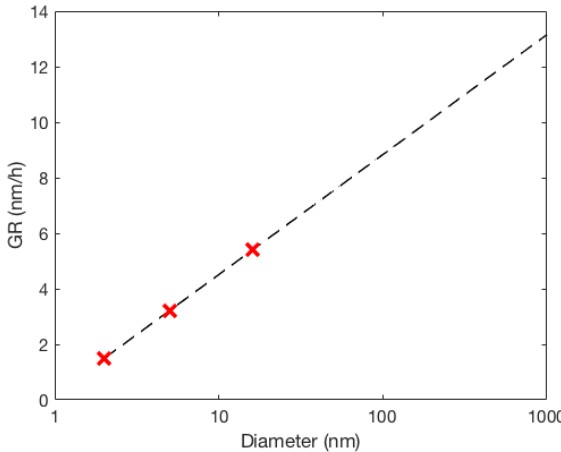

**Figure A6: Particle GR as a function of particle diameter. The red crosses show measured median values based on Zhou et al. (2020) and the black line is a fit to the measured values.**

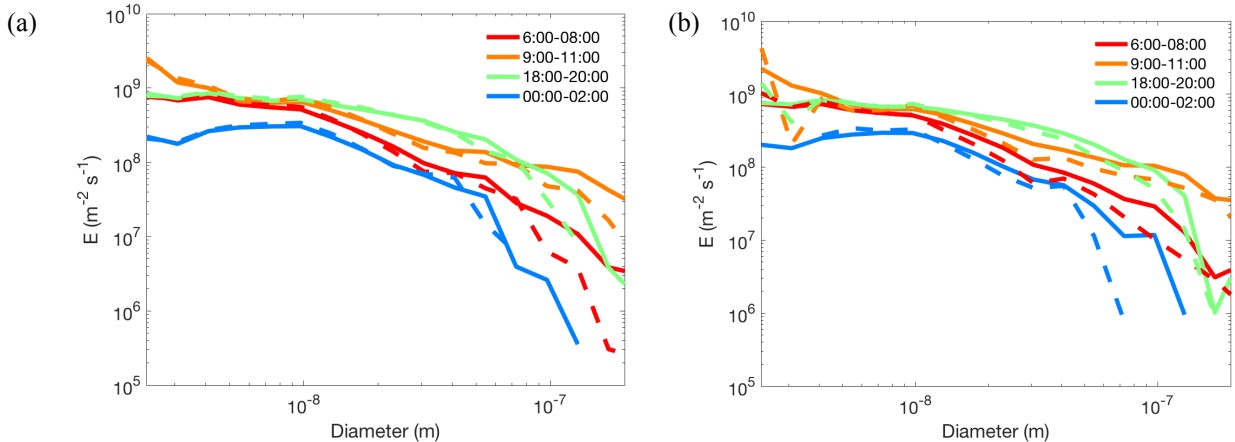

**Figure A7: Average particle number emission size distributions for non-NPF event days assuming (a) GR = 3 nm/h (solid lines) and GR that increases with size (dashed lines), (b) GR = 1.5 nm/h (solid lines) and GR = 6 nm/h (dashed lines).**

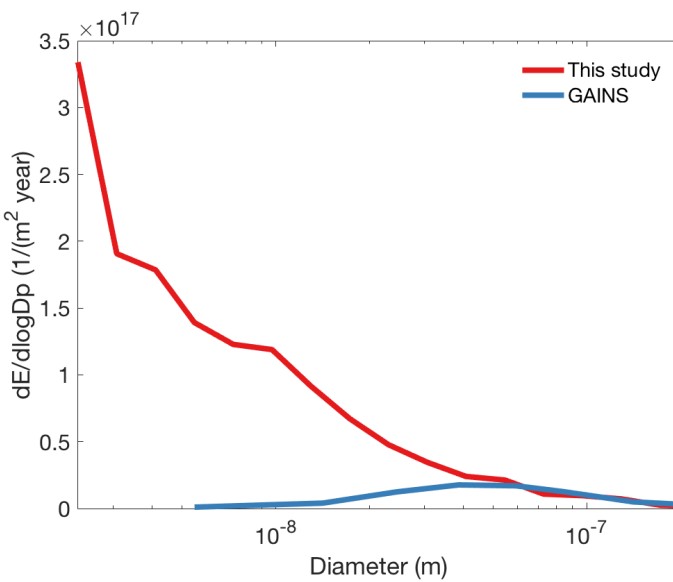


**Figure A8. Annual sum of particle number emissions at different sizes (normalized with the width of each size bin) based on particle number emissions calculated for non-NPF event days in this study (red line) and the GAINS model (blue line).**

### Data availability

The diurnally averaged particle number emissions are available at https://doi.org/10.5281/zenodo.3999054. GAINS emissions

are available at https://www.iiasa.ac.at/web/home/research/researchPrograms/air/PN.html. Supporting data are available from the authors upon request.

### Author contributions

JK and PP designed and developed the method, which was conceptualized by PP. CD, YF, YZ, TVK, ZL, YL, YW, LD, KRD, CY and MK contributed to data collection. JK performed the data-analysis. JK, PP, LD, JC, KRD, SH, JJ, TK, and MK

participated in the scientific discussion. JK prepared the manuscript with contributions from other authors. All the authors reviewed the manuscript.

### Competing interests

The authors declare that they have no conflict of interest.

## Acknowledgements

This work was supported by Academy of Finland (grant nos. 316114, 307331, and 311932), European Research Council (ATM-GTP; grant no. 742206), National Key R&D Program of China (2017YFC0209503) and National Natural Science Foundation of China (21876094).

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
