# Peer review of "Size-resolved particle number emissions in Beijing determined from measured particle size distributions"

_Atmospheric Chemistry and Physics, 2020_

## Referee Comment (RC1) · Anonymous Referee #1 · 29 Apr 2020

Summary

I recommend major revisions to the current paper, which concern mainly the explanation of the method and the validity of the method.

The paper introduces a very interesting method concept, where one is able to estimate size dependent number emission factors for an urban arae without having to perform specialized emission factor measurements, which require a larger infrastructure to accomplish. Hence, this is a cost-effective method.

However, I raise several critical questions, for which I would need answers before I am convinced that the method really works. The terms in equation (2) are not straightfor-

ward to comprehend. If authors are able to satisfactory examplify some of the terms or assumptions, it is likely that the method becomes more trustworthy. I outline these issues in the major remarks section below.

The introduction provides a very interesting input into the field of particle number size distribution emissions in urban areas, and has been very clearly presented. The result section is very well written and easy to understand and conclusions summarize the important findings well.

Major remarks

Chapter 2.2.1. How is it possible that the transport assumption is evened out? The quantity of the transported pollution to the urban area either depletes the particle concentration or enhances it in total after integrating it over all wind directions. Hence, I suspect there will be systematic bias. For example, if more particles are transported to the measurement site than what was there from the beginning (surrounding area contains more particles than emitted at the measurement site), there is a systematic positive bias of the emission factor, $E_i$ in equation (2). And vice versa if surrounding area has lower concentration of particles. And, there is a time aspect as well: high wind speed means in practice that the concentration that you experience at your urban area starts to resemble more and more the concentration of the long range transported aerosol rather than the urban emissions, since the urban emissions are quickly transported away from the urban area. The section 3.5.1 did not help to understand this issue. I think if the authors can show a concrete example of how this works in reality, it can be made trustworthy.

Chapter 2.2.2. In my view there are two terms involving the mixing vertically, the MLH parameter, and the $N_i*dLMH/dt$ parameter. A higher MLH means pollutants spread out over a larger height, meaning lower $dN_i/dt$ and vice versa. Hence, the vertical dilution is alraedy accounted for. So, why do you need the extra parameter $dLMH/dt$, which also relates to vertical dilution? How do the two parameters $dN_i/dt * MLH$ and $N_i * dLMH/dt$

relate to each other? I get your point with the increase in the boundary layer height in the morning causing dilution. But, this goes for both of these parameters. I also get the point in Figure 1, but it does not help me to understand.

What is the foot print area of the method in equation (2)? Traffic from a road more than 500 m away influences the concentration, which makes me believe that the foot print area is rather large? Is it km, or tens of km? But, it does not seem to be the entire urban area, since the emissions of traffic particles is higher from the method than for the entire urban area obtained with the GAINS model as discussed in section 3.6? Second: Is there a way to estimate a value of the size of the foot print area? Or how quickly the influence of urban emissions decrease in this method as function of distance from the measurement site?

Isn't the method rather impractical if you do not know the number of vehicles per hour in your foot print area? I mean, a higher traffic count gives a higher emission factor, Ei (per m squared and s). So, if you do not know the traffic count, you basically do not know why Ei is high. Is it because of high traffic emissions or just high traffic count? So, this method is only useful if you know the average traffic count in the area, meaning you can transform the emission factors to a useful quantity, emissions per m squared and s as function of number of vehicles in the foot print area. Then, with the number of vehicles in other parts of the city, you can get an emission factor for the entire city. But, you can not do that if you don't know the number of vehicles in your foot print area where you measured your emisson factor, Ei, or if you do not know the size of your foot print area.

Specific comment

Lines 55-57. Was the NPF apportioned to the secondary aerosol sources in Cai et al. (2020), or is it considerend as a completely seperate source? Cai et al. (2020) indicate that secondary sources, traffic and cooking together account for 100 % of the particles above 20 nm diameter, leaving no room for NPF as seperate source? This needs to be

explained in relation to the Cai et al. reference.

---

## Referee Comment (RC2) · Anonymous Referee #2 · 6 Jun 2020

In this study, a novel method was introduced for estimating size-resolved particle number emissions. The population balance method was used to estimate particle number emissions into a column extending from the ground to the top of the atmospheric mixing layer. In general, the manuscript is well written and provided some interesting results. But the origins and sources affecting particles number are very complex, but the variables represented in equation 2 is a bit insufficient. And the uncertainty of each variable in the equation needs to be considered more carefully. Therefore, I recommend it can be published on ACP after addressing the following comments.

1. The effect of advection transportation from surroundings was parameterized by the

experience of long-term measurements, which might induce large bias. The emissions of particles from surroundings are time-dependent, that should have diurnal and seasonal variation. In case using an annual-averaged parameter, these variations will be ignored. Therefore, I suggest considering the season cycle and diurnal cycle of advection transportation.

2. The height of the mixing layer was hard to estimate. In Eq. (2) we assume that ML is homogeneously mixed, which is not necessarily true in an urban environment. And the effect of dilution on their concentrations inside the ML might be overestimated.

3. Particle lossïijŽa constant deposition rate can cause uncertainties in estimated emissions. It should also have seasonal and diurnal variations.

4. The growth rate is assumed as a constant for all the size bines, 3 nm/h. From chapter 3.5, the particle emissions at small sizes are sensitive to the value of GR. If GR was considered as a constant, at non-NPF days, the second and the third term in equation (2) can be offset.

5. This method doesn't work well especially on NPF days. The parameters like J, GR, as well as start time may have big differences in different events, so these parameters or constants in the balanced equation should be changed in different episodes. Thus the daily averaged could not describe the progress of an NPF event. There are some questions: how to define the value of J and GR on non-NPF days? Did the authors use constant values of GR for all NPF days, like 3nm/h? If the nucleation processes are not considered on non-NPF days, how to get rid of the influence of nuclei particles on non-NPF days. I suggest, at least, to do some sensitivity test on the influence of NPF on the calculation.

6. The data of wind directions are used to show the affection of different origins of anthropogenic emissions, however, in my opinion, the map of the city should be added and other metrological conditions should also be considered.

7. There are few other data to support the prediction of the sources of particles. In addition, I think the impact of seasonal variation of sources to the particle concentration is not considered using one year's observation. For instance, the heating in winter in NCP area should discharge a large number of soot particles, but the paper didn't mention it.

———————————————————

---

## Author Comment (AC1) · 3 Jul 2020

**REPLIES TO REFEREES**

We thank the referees for their insightful comments and suggestions that improved our manuscript. We have answered to each of the referee's comments below. The text in **bold** is quoted from the referees' comments, the text in *italics* is quoted from the manuscript and the text highlighted in yellow has been added/modified in the revised manuscript.

Note that in addition to making the changes according to referees' suggestions, we also changed Figs 4-9 to new versions that are slightly different from the figures in the ACPD manuscript. The reason for this change is that we noticed that some of the days were missing from the data set used for calculating average emissions. However, the figures changed so little that the description of the figures in the text had to be changed only slightly in Sect. 3.3, and in the conclusions where the average contributions of different size ranges to the total emissions are given (these changes are also highlighted in yellow).

**Reply to Referee #1**

**Summary**

**I recommend major revisions to the current paper, which concern mainly the explanation of the method and the validity of the method. The paper introduces a very interesting method concept, where one is able to estimate size dependent number emission factors for an urban area without having to perform specialized emission factor measurements, which require a larger infrastructure to accomplish. Hence, this is a cost-effective method. However, I raise several critical questions, for which I would need answers before I am convinced that the method really works. The terms in equation (2) are not straightforward to comprehend. If authors are able to satisfactory exemplify some of the terms or assumptions, it is likely that the method becomes more trustworthy. I outline these issues in the major remarks section below. The introduction provides a very interesting input into the field of particle number size distribution emissions in urban areas, and has been very clearly presented. The result section is very well written and easy to understand and conclusions summarize the important findings well.**

We thank the referee for the helpful comments. It is true that some aspects of the method were not discussed in detail enough, especially related to effects of particle transport on the calculated emissions. We improved this in the revised manuscript following the referees' suggestions. Overall, we want to specify that currently our method cannot be considered to be able to determine particle number emissions accurately. This is due to the assumptions of the method discussed in Sect. 2.2 of the manuscript and in the answers below. In the future, we will develop the method by modeling the transport of particles from different pollution sources, but it is outside the scope of this study.

**Major remarks**

**Chapter 2.2.1. How is it possible that the transport assumption is evened out? The quantity of the transported pollution to the urban area either depletes the particle concentration or enhances it in total after integrating it over all wind directions. Hence, I suspect there will be systematic bias. For example, if more particles are transported to the measurement site than what was there from the beginning (surrounding area contains more particles than emitted at the measurement site), there is a systematic positive bias of the emission factor, $E_i$ in equation (2). And vice versa if surrounding area has lower concentration of particles. And, there is a time aspect as well: high wind speed means in practice that the concentration that you experience at your urban area starts to resemble more and more the concentration of the long range transported aerosol rather than the urban emissions, since the urban emissions are quickly transported away from the urban area. The section 3.5.1 did not help to understand this issue. I think if the authors can show a concrete example of how this works in reality, it can be made trustworthy.**

We agree that the effects of particle transport on the calculated emissions were not discussed clearly enough in the manuscript. We discuss these effects below (see also the answer related to the footprint area of the emissions), and we also improved their description in the manuscript.

First of all, our assumption is that emissions estimated by our method mainly represent emissions from sources that are constantly present in the urban region around our site and can be expected to be evenly distributed. Transport of particles to our site within this urban region should not cause significant bias in the estimated emissions if the assumption about nearly homogeneous emissions is valid. During the transport, particles grow in size, but this we correct for in Eq. (2). This can be understood by comparing Fig. 3a (the daily evolution of particle concentrations at different sizes), where we can follow the growth of ~10 nm particles emitted from traffic, and Fig. 4a (the daily evolution of particle emissions at different sizes) where we see a maximum in the emissions around 10 nm. However, the comparison of the emissions with different wind directions in Sect. 3.5.1 shows that wind direction also affects the calculated emissions of nucleation mode (< 30 nm) particles, which mainly originate within the urban region. For the particles in the smallest studied sizes (< ~ 6 nm) this likely results from the fact that northern wind directions favor NPF. But the observation that emissions for particles between 6 and 30 nm are higher for southern wind directions in the morning indicates that the assumption about homogeneous emissions within the urban region is not entirely correct. This is understandable as there are two busy roads located close to the measurement site. Still, the difference in the emissions with different wind directions is at most a factor of ~2, suggesting that our assumption is justified, when considering the overall uncertainties of the method.

On the other hand, transport of particles larger than ~30 nm from outside the urban region can cause bias in the calculated emissions, depending on the following issues:
1) Is there a region with rather homogeneous sources surrounding the urban region, or can the emission sources outside the urban region be considered as point sources located in different directions from our site?
2) Does wind direction have a strong diurnal cycle?

If the answer to both of the questions is no (i.e. the sources can be considered as point sources and the wind direction does not have a strong diurnal cycle), there should not be a significant bias caused by particle transport. This is because the transport of particles to the site has both positive and negative impacts on emissions (when particle concentrations increase due to transport they cause positive emissions and when they decrease they cause negative emissions). In case of point sources without a strong diurnal cycle in wind direction, these positive and negative contributions occur irregularly at different times, and thus when we average particle emissions over long enough data set, the effect of transport from these sources should be only minor. This can be illustrated by looking at the time evolution of particle concentrations and particle emissions for one day (Fig. R1). In the emission figure, we see positive emissions for particles with diameters of ~100 nm around 16:00, while their emissions become for some time negative before and after that. This is likely caused by transport of particles from some source. We also see clearly positive emissions for < 30 nm particles in the morning and in the evening, likely linked to traffic. When we look at the average daily evolution of particle number emissions (Fig. 4a and Fig. A2 of the revised manuscript), we can see that emissions for particles around 100 nm at 16:00 are low, because the effect of transport has been evened out by averaging over many days. However, we observe emissions from traffic, especially for sub-30 nm particles, as they occur at the same time every day all around our measurement site.

[Figure]

Figure R1. The daily evolution of particle number concentrations (left) and the calculated particle number emissions (right) on 8 November 2018.

If the answer to one of the questions above is yes (i.e. there is a region with rather homogeneous sources surrounding the urban region and/or there is a strong diurnal cycle in wind direction), there can be bias in the calculated emissions due to transport of particles. However, there are several indications that this bias is minor:

1) Average particle emissions are generally low for particles larger than 100 nm (see e.g. Fig A2 and A3 in the revised Appendix), for which the effect of transport should be most significant due to their long lifetime.
2) Emissions calculated with our method and those obtained from GAINS model agree very well for sizes above 60 nm (see Fig. 9 in the revised manuscript)
3) Our analysis related to the footprint area of emissions below suggests that most of the emissions originate within the area of the order of tens of km to different directions from our site.
4) The difference in the northern and southern wind directions is generally less than a factor of 2.

In the revised manuscript, we clarified the possible effects of particle transport in Sect. 2.2.1.

The comment on the effect of wind speed on calculated emissions is also relevant. When considering transport of particles within the urban region with rather homogenous emissions, wind speed only affects how far the particles observed at our site originate, and thus it should not significantly impact the calculated emissions (although when wind speed is high, dilution of particle concentrations due to mixing can be more efficient). However, assessing how wind speed affects calculated emissions when particles are transported outside the urban region is not straightforward but it depends for example on the following issues:

1) Is there a clear boundary after which emissions drop when going further away from our measurement site (e.g. outside the urban region)?
2) Does dilution rate of particles clearly increase with increasing wind speed?

If the answer to the first of these questions is yes, it means that when wind speed is higher, particles travel faster over the region with high emissions (e.g. the urban region) and thus particle concentrations have less time to accumulate in the air mass. As a result, we observe at high wind speeds lower particle concentrations at our site, leading to lower estimated particle emissions. If the answer to the first question is no (i.e. there are high emissions also far away from our measurement site), then emissions should not depend strongly on the wind speed, as wind speed only affects how far from our site observed particles have been emitted. If the answer to the second question is yes, then with high wind speeds particle concentrations from sources further away are more efficiently diluted before they arrive at our measurement site, and thus the transport of particles from far away sources does not impact the calculated emissions significantly. Regardless of the answers to these questions, in case of low wind speed, we can expect to see mainly local emissions from the urban region surrounding our measurement site.

To investigate the effect of wind speed on calculated particle emissions, we determined the average particle number emission size distributions for days when wind speed was predominantly over 1.1 m/s (70th percentile of 1 h averages of wind speed; 20 days) and for days when wind speed was predominantly below 0.6 m/s (30th percentile of 1 h averages of wind speed; 10 days). The results of this analysis are shown in Fig. R2. Figure R2 shows that the differences in the emissions between different wind speeds are generally minor. During the

day, the ratio between emissions at low and high wind speeds vary between 0.6 and 1.3 at different sizes. In the evening and at night, the difference is higher; for example, at 18:00-20:00 the emissions for particles above 30 nm are higher by up a factor of ~2.5 with low wind speed. This can be caused by accumulation of particles to the air mass over the urban region. The fact that this effect is clearest in the evening and at night may be connected to a more stable boundary layer at that time. On the other hand, the emissions for the smallest particles are higher at higher wind speeds, which is likely connected to atmospheric cluster formation. Still, when considering the overall sources of uncertainty in our calculations and the fact that the average particle emission size distributions for two wind speed cases are calculated from a very limited number of days, the differences between different wind speeds can be considered relatively minor.

[Figure]

Figure R2. Average particle number emission size distributions for non-NPF event days when wind speed is predominantly below 0.6 m/s (solid lines) and over 1.1 m/s (dashed lines). The left panel shows the figure on a linear scale and the right panel on a logarithmic scale.

In the revised manuscript we added Figure R2 and discussion about it to Sect. 3.5.1. We also improved the description of the effects of particle transport on calculated emissions in this section and in the conclusions.

Overall, to evaluate the effect of particle transport on the calculated emissions more quantitively would require modeling of the transport of particles from different sources to our measurement site under different meteorological conditions (e.g. wind speed). This will be a topic of a future study. Still, based on the analysis presented in these review answers and in the manuscript, we feel confident that also at its current form, our method can provide knowledge of particle number emissions in urban environments which is of value to the scientific community. In the revised manuscript, we now mention the need for the further improvement of our method in the abstract, in the end of Sect. 3.5.1 and in the conclusions.

**Chapter 2.2.2. In my view there are two terms involving the mixing vertically, the MLH parameter, and the $N_i$*dMLH/dt parameter. A higher MLH means pollutants spread out over a larger height, meaning lower $dN_i$/dt and vice versa. Hence, the vertical dilution is already accounted for. So, why do you need the extra parameter dMLH/dt, which also relates to vertical dilution? How do the two parameters $dN_i$/dt * MLH and $N_i$ * dMLH/dt relate to each other? I get your point with the increase in the boundary layer height in the morning causing dilution. But, this goes for both of these parameters. I also get the point in Figure 1, but it does not help me to understand.**

The two terms come from the derivative of column particle number concentration ($N_i \times$ MLH). The column number concentration at a certain point in time can change due to emissions and other processes affecting the number concentration measured at the ground level; this is described by the term $dN_i$/dt $\times$ MLH. In addition, the column number concentration can at that point in time change due to increase of MLH, causing a decrease in particle concentration at the ground level. This is described by the second term $N_i \times$ dMLH/dt.
This is now clarified in Sect. 2.1 of the revised manuscript:

*The time derivative of the column number concentration can be divided to two terms: the first one is $\frac{dN_i}{dt} \times MLH$, which describes the change of the column particle number concentration due to processes affecting directly particle number concentration $N_i$, and the second term is $N_i \frac{dMLH}{dt}$, which describes the dilution of the concentration $N_i$, due to increase of mixing layer height (MLH) in the morning.*

**What is the foot print area of the method in equation (2)? Traffic from a road more than 500 m away influences the concentration, which makes me believe that the footprint area is rather large? Is it km, or tens of km? But, it does not seem to be the entire urban area, since the emissions of traffic particles is higher from the method than for the entire urban area obtained with the GAINS model as discussed in section 3.6? Second: Is there a way to estimate a value of the size of the foot print area? Or how quickly the influence of urban emissions decreases in this method as function of distance from the measurement site?**

Thanks for the good comment. As explained above, determining exactly how transport of the particles from sources at different distances and directions from our site affects the calculated emissions would require modeling work, which is outside the scope of this study. However, we can get a ballpark estimate for the area which mainly contributes to calculated emissions by looking at the difference between emissions calculated for different wind directions (Fig. 7 in the revised manuscript) and comparing it to the maps of population density (Fig. R3) and emissions of $PM_{2.5}$, $NO_x$, CO and $SO_2$ obtained from emission inventories (Fig. R4). First of all, as discussed in Sect. 3.5.1 and shown in Fig. 7, the differences in the emissions between north-western and south-eastern wind directions are relatively minor. At the smallest sizes, particle production is higher for north-western directions but this is likely linked to atmospheric cluster formation. Apart from that, the clearest difference between wind directions is observed in the morning (06:00-08:00) and at night (00:00-02:00) for particles between 7 and 100 nm, for which the emissions are higher by a factor of ~1.4–2 for south-eastern directions. This can be because the area with high emissions extends further in that direction. On the other hand, at 18:00-20:00 the emissions for particles between 10 and 50 nm are slightly higher, by up to a factor of 1.6, with north-eastern wind directions. When looking at the population density and the emissions of trace gases in the region surrounding our measurement site (Figs R3 and R4), a strong decline in particle emissions can be expected ~20 km west and ~50 km north of our site, and a moderate, more gradual, decline ~100-200 km east and ~50 km south of the site. The fact that the difference in the calculated particle emissions between north-western and south-eastern directions is at most a factor of 2 indicates that most of the emissions determined with our method originate within a radius of a few tens of km from our site, within urban Beijing. Still, one should note that there are two busy roads located close to the measurement site, which can be expected to enhance the calculated emissions relative to the average emissions of the urban region.

[Figure]

Figure R3. The maps of (a) urban Beijing and its main roads, and (b) the region around Beijing with the population density shown as color. The location of the measurement site of BUCT is shown with a magenta cross in both maps. The green rectangle in (b) corresponds to the region shown in (a).

[Figure]

Figure R4. Annual emissions of (a) PM2.5, (b) NOx, (c) CO and (d) SO2 for the year 2010 based on the MIX emission inventory (Li et al., 2017) in the region around Beijing.

In the revised manuscript, we added the maps of urban Beijing and the population density in Sect. 2.3, and the maps with emissions of $PM_{2.5}$ and trace gases in Appendix.

Regarding the comparison between our results and the GAINS model, we would like to point out that GAINS is known to underestimate the emissions for the smallest sizes (see Paasonen et al., 2010), because the emission factors for many sources include only particles that are nonvolatile and/or particles larger than 10 nm. For this reason, the difference between calculated emissions and GAINS for the smallest particle sizes indicates the limitations of the GAINS model in describing the emissions of the smallest particles. This is stated in the end of Sect. 3.6 and in the conclusions. However, the fact that our method and GAINS agree very well for particles larger than 60 nm, suggests that the footprint area of our method can be close to the same area as the chosen grid size in GAINS, i.e. 50 km × 50 km. This is consistent with our estimation of the footprint area above. In the revised manuscript, we mention this in the end of Sect. 3.6:

*It also suggests that the emissions calculated with our method are mainly influenced by emissions in the area approximately of the same size as the chosen grid size of GAINS, i.e. $50 \times 50$ km², which is consistent with our estimation in Sect. 3.5.1.*

To summarize, regarding the effect of particle transport, we modified the manuscript followingly:
1) We emphasize in the revised manuscript that due to different sources of uncertainty, the emissions estimated by our method should not be considered exact. We also explain that we will perform modeling to improve the method in the future.
2) We improved Sect. 2.2.1 discussing the uncertainties of our method related to the effect of transport.
3) We added the maps of Beijing and the population density (Fig. R3) in Sect. 2.3, a figure of emissions with different wind speeds (Fig. R2) in Sect. 3.5.1 and the emission inventory maps (Fig. R4) in Appendix. In

Sect. 3.5.1 and in the conclusions, we now discuss in more detail the effect of transport on calculated emissions utilizing the new figures.

**Isn't the method rather impractical if you do not know the number of vehicles per hour in your foot print area? I mean, a higher traffic count gives a higher emission factor, $E_i$ (per m squared and s). So, if you do not know the traffic count, you basically do not know why $E_i$ is high. Is it because of high traffic emissions or just high traffic count? So, this method is only useful if you know the average traffic count in the area, meaning you can transform the emission factors to a useful quantity, emissions per m squared and s as function of number of vehicles in the foot print area. Then, with the number of vehicles in other parts of the city, you can get an emission factor for the entire city. But, you cannot do that if you don't know the number of vehicles in your foot print area where you measured your emission factor, $E_i$, or if you do not know the size of your footprint area.**

It is true that the emissions determined with our method cannot be directly applied for solving whether the emissions are high due to high activity levels (e.g. high traffic count) or high emission factors (e.g. vehicles with high emissions). However, our method provides comparison data for the emissions calculated from the traffic count and emission factors for example using a model like GAINS, or other models with higher spatial and temporal resolution of activity levels. We also want to note that our aim with this method is not to determine only vehicle emissions, but to obtain a general estimate for particle number emissions in urban Beijing originating from different sources. Still, it can be useful to compare the emissions estimated with our method to traffic rates of different types of vehicles, or traffic emissions obtained with some other method. In Sect. 3.3 we compare particle number emissions from our method with $PM_{2.5}$ emissions estimated for gasoline and diesel vehicles with EMBEV-Link (Link-level Emission factor Model for the BEijing Vehicle fleet) model by Cai et al. (2020). Extending this analysis further would be interesting, but it is outside the scope of this study.

**Specific comment**

**Lines 55-57. Was the NPF apportioned to the secondary aerosol sources in Cai et al. (2020), or is it considered as a completely separate source? Cai et al. (2020) indicate that secondary sources, traffic and cooking together account for 100 % of the particles above 20 nm diameter, leaving no room for NPF as separate source? This needs to be explained in relation to the Cai et al. reference**

Cai et al. (2020) performed the PMF analysis only for days without NPF events, and thus NPF is not included as a source in their PMF results. It should also be noted that their data set covered only months from April to July, which can explain the missing of biomass burning and coal combustion as identified sources. This is now clarified in the revised version of the manuscript:

*They used data from April to July 2018, excluding NPF event days from the analysis.*

**Reply to Referee #2**

**In this study, a novel method was introduced for estimating size-resolved particle number emissions. The population balance method was used to estimate particle number emissions into a column extending from the ground to the top of the atmospheric mixing layer. In general, the manuscript is well written and provided some interesting results. But the origins and sources affecting particles number are very complex, but the variables represented in equation 2 is a bit insufficient. And the uncertainty of each variable in the equation needs to be considered more carefully. Therefore, I recommend it can be published on ACP after addressing the following comments.**

We thank the referee for the constructive comments. It is true that particle dynamics in urban environments is very complex, and that is why we need to make many simplifications in our method. As explained also in the reply to Referee #1, the emissions estimated with our method should not be considered precise due to these

simplifications and other sources of uncertainty. We now state this more clearly in the revised manuscript and we added there more discussion on different sources of uncertainty.

**1.The effect of advection transportation from surroundings was parameterized by the experience of long-term measurements, which might induce large bias. The emissions of particles from surroundings are time-dependent, that should have diurnal and seasonal variation. In case using an annual-averaged parameter, these variations will be ignored. Therefore, I suggest considering the season cycle and diurnal cycle of advection transportation.**

We would like to clarify that we are not considering advection in our equation. As also explained in the answers to Referee #1, our assumption is that by averaging over long enough data set, the effect of particle advection from sources further away from our site is only minor. This is now clarified in the manuscript.

However, it is true that turning of the wind direction during the day (see Fig. A4) can cause some bias in the average diurnal cycle of calculated particle number emissions. Still, this bias does not seem to be major due to the following reasons mentioned also in the answers to Referee #1:

1)  Average particle emissions are generally low for sizes above 100 nm (see e.g. Fig A2 and A3 in the revised Appendix), for which transport from outside the urban region should affect most.
2)  Emissions calculated with our method and those obtained from GAINS model agree very well for sizes above 60 nm (see Fig. 9 in the revised manuscript).
3)  Our analysis related to footprint area of emissions suggests that most of the emissions calculated with our method originate within some tens of km from our measurement site.
4)  The difference in particle number emissions between the northern and southern wind directions is mostly below a factor of 2, which is a rather good agreement considering the overall uncertainties of our method.

We do not consider seasonal variation of the emissions in this study, because of the limitations of our data set. Although our measurements are from over one-year period, the final corrected data set included only 136 days of which 76 days were non-NPF event days. These days covered months from October to May. This is now clarified in the revised manuscript as follows:

==The final corrected data set includes 136 days of particle size distributions between 1 nm and 10 μm, covering months from October to May==

**2. The height of the mixing layer was hard to estimate. In Eq. (2) we assume that ML is homogeneously mixed, which is not necessarily true in an urban environment. And the effect of dilution on their concentrations inside the ML might be overestimated.**

It is true that ML is not necessarily homogeneously mixed in an urban environment. We discuss this in Sect 2.2.2 of the manuscript, referring to previous studies indicating that ML is Beijing is not always well-mixed and stating that this may cause us to over- or underestimate particle emissions. Unfortunately, no measurements of the vertical profile of ML were available during our measurement period. Therefore, using the current data set, we are not able to consider inhomogeneous mixing of ML in our emission calculations. We agree that more knowledge of the effects of urban boundary layer development on particle dynamics in Beijing would be needed and therefore we mention this in the conclusions of the revised manuscript:

*To improve the method, more knowledge of particle dynamics in urban environments is needed, such as the loss rates of different sized particles due to evaporation and deposition and ==the impacts of the urban boundary layer development on particle dynamics==.*

**3. Particle lossïijŽa constant deposition rate can cause uncertainties in estimated emissions. It should also have seasonal and diurnal variations.**

We agree that it is quite a crude simplification to assume a constant deposition rate for all particle sizes. However, considering deposition more realistically would require information on the roughness of the surface and stability of the boundary layer (dry deposition; Zhang and Wexler, 2002) and the size distribution of rain droplets, rain intensity and collision efficiency (wet deposition; Laakso et al., 2003). Therefore, we decided to

assume a constant deposition rate. However, this assumption affects significantly only the emissions of particles larger than 100 nm, for which coagulation losses are low. We now further clarified these issues in Sect. 2.2.3 where the assumption about deposition rate is discussed:

*In addition, when we describe the removal of particles by deposition, we assume a constant deposition rate for all particle sizes, corresponding to the lifetime of 1 week (Stocker, et al., 2013). In reality, dry and wet deposition are size- and time-dependent processes, which depend for example on the properties of available surfaces, boundary layer and rainfall (e.g. Laakso et al., 2003; Zhang and Wexler, 2002). Thus, a constant deposition rate can cause uncertainties in estimated emissions, especially for the largest particles for which deposition is most important due to low coagulation losses. With our assumption for the deposition rate, deposition affects significantly only the emissions of particles larger than 100 nm, by increasing their emissions by maximum of ~20% at night and less during the day.*

Note that we also mention in the conclusions that to develop our method further, more knowledge of the loss rates of different-sized particles due to deposition is needed.

**The growth rate is assumed as a constant for all the size bins, 3 nm/h. From chapter 3.5, the particle emissions at small sizes are sensitive to the value of GR. If GR was considered as a constant, at non-NPF days, the second and the third term in equation (2) can be offset.**

It is true that GR affects the emissions at the smallest size. This is because the terms describing the growth into and out of the size bin do not offset each other when particle concentration changes strongly with size. In the revised manuscript, we explain this in Sect. 2.2.4:

*With a constant GR, the terms in Eq. (2) describing the growth into and out of the size bin offset each other if particle concentration does not significantly change with size.*

Furthermore, for the smallest studied size bin, the growth into the size bin term is omitted to include the contribution of atmospheric clustering to particle production (see the end of Sect. 2.1), and thus GR affects most the smallest size bin. This applies also to non-NPF event days, because weak atmospheric cluster formation occurs also then. This is mentioned in Sect. 3.5.2. As also explained in Sect. 3.5.2, we chose to use a constant value of 3 nm/h for GR because the results are only sensitive to GR at the smallest sizes, and according to Zhou et al. (2020), GR = 3 nm/h is a good approximation at these sizes.

**5. This method doesn't work well especially on NPF days. The parameters like J, GR, as well as start time may have big differences in different events, so these parameters or constants in the balanced equation should be changed in different episodes. Thus the daily averaged could not describe the progress of an NPF event. There are some questions: how to define the value of J and GR on non-NPF days? Did the authors use constant values of GR for all NPF days, like 3nm/h? If the nucleation processes are not considered on non-NPF days, how to get rid of the influence of nuclei particles on non-NPF days. I suggest, at least, to do some sensitivity test on the influence of NPF on the calculation.**

We agree that our method does not describe the particle dynamics on NPF event days accurately. One reason for this is that not all sub-6 nm particles formed by NPF are observed to grow to larger sizes (see Sect. 3.2), which may be connected to inhomogeneities in the area where NPF occurs around our measurement site. In the future, we hope to develop our method to better describe the particle dynamics on NPF event days, as mentioned in the end of the conclusions.

However, we would like to point out that we do not have J in our equations. The effect of NPF is included in Eq. (2) with the term describing the growth into the smallest size bin (see Sect. 2.1). As discussed for example in Sect. 2.2, the contribution of atmospheric cluster formation to particle production in the smallest size bin can be seen also on non-NPF event days. However, on non-NPF event days the growth of the smallest particles to larger sizes does not occur on a large enough area that we could observe it at our measurement site similar to the growth of particles in a regional NPF event. Still, it is reasonable to assume that particles grow also on

non-NPF event days, because we observe the growth of particles emitted by traffic on non-event days (see Fig. 3a of the revised manuscript).

As discussed in the manuscript as well as answers above, we used a constant value of GR for both NPF event days and non-event days. We also studied the effect of the value of GR on calculated emissions in Sect. 3.5.2, and found that our results are sensitive to GR only at the smallest sizes where GR = 3 nm/h is a good assumption.

**6. The data of wind directions are used to show the affection of different origins of anthropogenic emissions, however, in my opinion, the map of the city should be added and other metrological conditions should also be considered.**

Thanks for the good suggestion. As explained in the answers to Referee #1, the revised manuscript now includes the maps of the city of Beijing and its surroundings, showing the main roads, population density and the location of our measurement site. In addition, we added maps showing the emissions of $PM_{2.5}$ and trace gases ($NO_x$, CO and $SO_2$) based on emission inventories in Appendix. Furthermore, in addition to discussing the effect of wind direction, we now consider the effect of wind speed on the calculated emissions in Sect. 3.5.1 (see the answers to Referee #1).

**7. There are few other data to support the prediction of the sources of particles. In addition, I think the impact of seasonal variation of sources to the particle concentration is not considered using one year's observation. For instance, the heating in winter in NCP area should discharge a large number of soot particles, but the paper didn't mention it.**

It is true that particle emissions in Beijing are expected to exhibit a seasonal cycle. However, in this paper we chose not to study the seasonal variation in particle emissions due to the limitations of our data set. Our data set covers only months from October to May, and thus the data is mostly from the heating season. The analysis of the seasonal cycle of particle number emissions will be performed in a future study, after a longer data set covering also summer months has been collected. In the revised manuscript, we clarify in the methods section, that our data set covers only months from October to May (see the answer to the first comment). In addition, we now mention the seasonal variation of particle emissions observed in previous studies in our introduction:

*The relative strength of these sources varies seasonally; for example, coal combustion is a significant source only during the residential heating period (Hu et al., 2017), which is usually between mid-November and mid-March.*

**References**

Li, M., Zhang, Q., Kurokawa, J. I., Woo, J. H., He, K., Lu, Z., Ohara, T., Song, Y., Streets, D. G., Carmichael, G. R., Cheng, Y., Hong, C., Huo, H., Jiang, X., Kang, S., Liu, F., Su, H. and Zheng, B.: MIX: A mosaic Asian anthropogenic emission inventory under the international collaboration framework of the MICS-Asia and HTAP, Atmos. Chem. Phys., 17(2), 935–963, doi:10.5194/acp-17-935-2017, 2017.

---

## Author Response (AR2)

**AUTHOR'S RESPONSE**

We have answered to the referee's comments below. The text in **bold** is quoted from the referee's comments, the text in *italics* is quoted from the manuscript and the text highlighted with yellow has been added to the revised manuscript. The revised manuscript with changes highlighted in yellow can be found after the replies.

**Reply to Referee #1**

**Thank you for the revision and answers to the review. I am very satisfied with all answers and the new version of the manuscript. I am willing to accept the paper subject to a minor correction, which concerns two points:**
**1. Please do not be so modest as to the positive benefits of your method. You mention more or less indirectly in the abstract and the introduction section that models like GAINS can be validated with experimental data from your method. Please state this very strong benefit of your method more directly in the introduction section.**
**2. By knowing the typical traffic activity in the footprint area, you can produce an emission factor, which is size dependent particle number size concentration per vehicle km driven of a typical vehicle in the fleet mix in the current town. This can be achieved during certain periods when you can assume car emissions dominate contribution to particle number concentration, or in other cities, where you know that traffic dominates the emissions in the city. You get this emission factor by simply dividing your emission values in number per square m and s with the traffic activity in vehicle km driven of the fleet per square m and s in the footprint area. An example of how this is done can be seen in for example Mårtensson et al. 2006 (Atmosheric Chemistry and Physics, 6, 769-785). This emission factor is invaluable as input data to for example dispersion models. Please also state this very strong benefit of your method in the introduction section very clearly. Additional data might even give you this emission factor seperated for light and heavy duty vehicles, which is also exemplified by Mårtensson et al., 2006.**

We thank the reviewer for the encouraging comments. In the revised manuscript we state the usefulness of our method for modelling purposes already in the abstract:

*Overall, our method is proven to be a useful tool for gaining new knowledge of the size distributions of particle number emissions in urban environments, and for validating emission inventories and models.*

We also now clearly mention in the introduction different modeling applications, for which the observation-based methods (like one presented in this study) can be used:

*Besides validating integrated assessment models, observation-based methods can be directly used to derive particle number emission factors for traffic (see e.g. Mårtensson et al., 2006), needed in different air quality modeling applications.*

In addition to these changes, we slightly modified one sentence and a citation in Sect. 3.4 to cite a very recent publication on NPF at this site, instead of a paper that we originally cited that is still under review:

[revised manuscript text omitted]